



# Critical scales to explain urban hydrological response

Elena Cristiano[1], Marie-claire ten Veldhuis[1], Santiago Gaitan[1,2], Susana Ochoa Rodriguez[3], and Nick van de Giesen[1]

[1]Department of Water Management, Delft University of Technology, Postbox 5048, 2600 GA, Delft, The Netherlands
[2]Environmental Analytics, Innovation Engine BeNeLux, IBM, Amsterdam, The Netherlands
[3]RPS Water, Derby, UK

*Correspondence to:* Elena Cristiano (E.Cristiano@tudelft.nl)

**Abstract.** Rainfall variability in space and time, in relation to catchment characteristics and model complexity, plays an important role in explaining the sensitivity of hydrological response in urban areas. In this work we present a new approach to classify rainfall variability in space and time and we use this classification to investigate rainfall aggregation effects on urban hydrological response. Nine rainfall events, measured with a Dual polarimetric X-Band radar at the CAESAR Site (Cabauw

Experimental Site for Atmospheric Research, NL), were aggregated in time and space in order to obtain different resolution combinations. The aim of this work was to investigate the influence that rainfall and catchment scales have on hydrological response in urban areas. Three dimensionless scaling factors were introduced to investigate the interactions between rainfall and catchment scale and rainfall input resolution in relation to the performance of the model. Results showed that (1) rainfall classification based on cluster identification well represents the storm core, (2) aggregation effects are stronger for rainfall

than flow, (3) model complexity does not have a strong influence compared to catchment and rainfall scales for this study case, (4) scaling factors allow to select the adequate rainfall resolution to obtain a given level of accuracy in the calculation of hydrological response.

## 1 Introduction

Rainfall variability in space and time influences the hydrological response, especially in urban areas, where hydrological

response is fast and flow peaks are high (Fabry et al., 1994; Faures et al., 1995; Smith et al., 2002; Emmanuel et al., 2012; Gires et al., 2012; Smith et al., 2012; Ochoa-Rodriguez et al., 2015; Thorndahl et al., 2017). Finding a proper match between rainfall resolution and hydrological model structure and complexity is important for reliable flow prediction (Berne et al., 2004; Ochoa-Rodriguez et al., 2015; Pina et al., 2016; Rafieeinasab et al., 2015; Yang et al., 2016). High resolution rainfall data are required to reduce errors in estimation of hydrological responses in small urban catchments (Niemczynowicz, 1988; Schilling,

1991; Berne et al., 2004; Bruni et al., 2015; Yang et al., 2016). New technologies and instruments have been developed in order to improve rainfall measurements and capture its spatial and temporal variability (Einfalt et al., 2004; Thorndahl et al., 2017). In particular, the development and use of weather radars for hydrological applications has increased in the last decades (Niemczynowicz, 1999; Krajewski and Smith, 2005; Leijnse et al., 2007; van de Beek et al., 2010; Otto and Russchenberg, 2011; Berne and Krajewski, 2013), improving the spatial resolution of rainfall data (Cristiano et al., 2017).



The increase of high-resolution topographical data availability leaded to a development of different types of hydrological models (Mayer, 1999; Fonstad et al., 2013; Tokarczyk et al., 2015). These models represent spatial variability of catchments in several ways, varying from lumped systems, where spatial variability is averaged into sub-catchments, to distributed models, which evaluate the variability dividing the basin with a mesh of interconnected elements based on elevation (Zoppou, 2000;

Fletcher et al., 2013; Pina et al., 2014; Salvadore et al., 2015). Salvadore et al. (2015) analysed the most used hydrological models, comparing different model complexities and approaches. An investigation of the differences between high resolution semi and fully distributed models was proposed by Pina et al. (2016), where flow patterns generated with different model types were studied and compared to observations. This work suggested that although fully distributed models allow to represent catchment variability in space in a more realistic way, they did not lead to the best modelling results because the operation of

this type of models requires very high quality and resolution data, including rainfall input.

Both rainfall and model resolution and scale are expected to have strong effects on hydrological response sensitivity. An increase of sensitivity is expected for small drainage areas and for rainfall events with high variability in space and time. Sensitivity to rainfall data resolution generally increases for smaller urban catchments. However, sensitivity of hydrological models at different rainfall and catchments scale still remains poorly understood. This work builds upon Ochoa-Rodriguez

et al. (2015), who showed that the influence of rainfall input resolution decreases with the increase catchment area and that the interaction between spatial and temporal rainfall resolution is quite strong. We investigate sensitivity of urban hydrological response to different rainfall and catchment scales, with the aim of answering the following research questions:

– How should rainfall variability in space and time be classified?

– How does small scale rainfall variability affect hydrological response in a highly urbanized area?

– How does model complexity affect sensitivity of model outcomes to rainfall variability?

– How does the relationship between storm scale and basin scale affect hydrological response?

The paper is structured as follows. Section 2 presents the case study, describing study area, models and rainfall data used in this work. Methodology applied to identify variability in space and time of model and rainfall and hydrological analysis are explained in Section 3. Section 4 presents the results connected to the model and rainfall variability analysis and to

the hydrological analysis respectively. In Section 5, results are discussed, by comparing the influence of rainfall and model characteristics and identifying dimensionless parameters to describe the relation between rainfall and model scale and rainfall resolution used. Conclusions and future steps are presented in the last section.



## 2 Pilot catchment and datasets

### 2.1 Study area and available models

The city of London (UK) is exposed to high pluvial flood risk in the last years. The Cranbrook catchment, in the London Borough of Redbridge, is a densely urbanized residential area. For this reason, it has been chosen as study area. A total area of
approximately 860 ha is connected to the drainage network, and rainfall is drained with a separate sewer system.

For this small catchment, several urban hydrodynamical models have been set up in InfoWorks ICM (Innovyze, 2014). Three models with different representations of surface spatial variability, are used in this study: SD1 - Simplified semi-distributed low resolution, SD2 - Semi-distributed high resolution and FD - Fully distributed 2D high resolution.

Table 1 summarises the main characteristics of the three models: number of nodes, pipes and sub-catchments, dimensions
of subacatchments, two dimensional surface elements and degree of imperviousness. The first model, SD1, is a low resolution semi-distributed model, initially setup by the water utility (Thames Water) back in 2010 to gain a strategic understanding of the catchment. This model divides the area into 51 sub-catchments, connected with 242 nodes and 270 pipes, for a total drainage network length of just over 15 km. The other two models, SD2 and FD, have been developed at Imperial College London (Simões et al., 2015; Wang et al., 2015; Ochoa-Rodriguez et al., 2015; Pina et al., 2016). SD2 and FD share the same sewer
network design(6963 nodes and 6993 pipes), but use different surface representations. In SD2 the drainage area is divided into 4409 sub-catchments, where rainfall runoff processes are modelled in a lumped way and wherein rainfall is assumed to be uniform. In FD, instead, the surface is modelled with a dense triangular mesh (over 100'000 elements), based on a high resolution (1 m x 1 m) Digital Terrain Model (DTM). The rainfall - runoff transformation is different for the two types of model. For SD2, runoff volumes are estimated from rainfall depending on the land use type and routed, while for FD, runoff
volumes are estimated and applied directly on the two-dimensional elements of the overland surface. Figure 1 illustrates how the surface area is modelled for each of the three models and sewer networks.

### 2.2 Rainfall data

Rainfall events were selected from a dataset collected by a dual polarimetric X-Band weather radar located in Cabauw (CAE-SAR weather station, NL). For technical specifications of the X-band radar device see Ochoa-Rodriguez et al. (2015). The
selected events were measured with a resolution of 100 m x 100 m in space and 1 min in time, much higher than what is obtained with conventional radar networks (1000 m x 1000 m and 5 min). Rainfall data where applied to the Cranbrook catchment, using sixteen combinations of space and time resolution aggregated from the 100 m - 1 min resolution: four spatial resolutions, $\Delta s$, (100 m, 500 m, 1000 m and 3000 m) with four temporal resolutions, $\Delta t$, (1 min, 3 min, 5 min and 10 min) (see Ochoa-Rodriguez et al. (2015) for a motivation of the different resolution combinations). Nine rainfall events, measured
between January 2011 and May 2014, were used as model input in this study. Storm characteristics are presented in Table 2.





## 3 Methods

### 3.1 Characterizing storms' spatial and temporal rainfall scale

In this section, different ways of classifying spatial and temporal rainfall scale are described. We propose a new characterization of spatial and temporal rainfall variability, based on percentage of coverage above selected thresholds.

### 3.1.1 Spatial rainfall scale based on climatological variogram

Ochoa-Rodriguez et al. (2015) presented the theoretical spatial rainfall resolution required for an hydrological model in urban area, deriving it starting from a climatological (semi-) variogram. The (semi-) variogram $\gamma$ was calculated at each time step as:

$$\gamma = \frac{1}{2n} \sum_{t}^{n} (R(x) - R(x+h))^2,  \tag{1}$$

where $n$ is the numbers of radar pixel pairs located at a distance $h$, $R$ is the rainfall rate and $x$ is the center of the given pixel, normalized by the sample variance and averaged over the time period. The obtained variogram, characteristic of the averaged rainfall spatial structure during the peak period, was then fitted with an exponential variogram and the area $A$ under the correlogram was calculated for the exponential variogram as: $A = \frac{2\pi r^2}{9}$. $A$ can be considered as the average area of spatial rainfall structure estimated with radar measurements over the study area (Ochoa-Rodriguez et al., 2015). Characteristic length scale $r_c$[L] of a rainfall event was defined as: $r_c = (\frac{\sqrt{2\pi}}{3})r$, where $r$ [L] is the variogram range. Minimum required spatial resolution $\Delta s_r$ was defined in this work as half of the storm characteristic length scale:

$$\Delta s_r = \frac{r_c}{2} \cong 0.418r.  \tag{2}$$

This parameter describes the spatial variability of the rainfall event core.

### 3.1.2 Rainfall Spatial Variability Index

Another parameter to quantify and compare the spatial variability of rainfall is the spatial rainfall variability index $I_\sigma$. This parameter was at first proposed by Smith et al. (2004), called index of rainfall variability, and then recently redefined by Lobligeois et al. (2014). This index was estimated as:

$$I_\sigma = \frac{\sum_t \sigma_t R_t}{\sum_t R_t}  \tag{3}$$

where $\sigma_t$ is the standard deviation of spatially distributed hourly rainfall across all pixels in basin, per time-step $t$, and $R_t$ represents the spatially averaged rainfall intensity per time step. As can be seen, $I_\sigma$ corresponds to a weighted average, based on instantaneous intensity, of the standard deviation of the rainfall field during a given storm event. Small values of $I_\sigma$ indicate a low rainfall variability, typical of stratiform rainfall events. Large values of $I_\sigma$ generally represent convective storms, characterized by high spatial variability. In the study presented by Lobligeois et al. (2014), $I_\sigma$ was applied to rainfall data measured in the French region with a resolution of 1000 m - 5 min and it was varying between 0 and 5.





### 3.1.3 Storm velocity and temporal rainfall variability based on storm cell tracking

Ochoa-Rodriguez et al. (2015) presented a characterization of rainfall velocity and a definition of the minimum required temporal resolution. Storm motion was defined applying the TREC method (TRacking Radar Echoes by Correlation) proposed by Rinehart and Garvey (1978) This method allows to obtain at each time step a vector representing velocity magnitude and direction. The minimum required temporal resolution $\Delta t_r$, was obtained considering time that a storm needs to pass over the storm event characteristic length scale $r_c$. $\Delta t_r$ can be written as:

$$\Delta t_r = \frac{r_c}{|\bar{v}|}, \tag{4}$$

where $|\bar{v}|$ [L t$^{-1}$] correspond to the mean storm velocity magnitude. $|\bar{v}|$ is obtained from the average of the velocity vectors, estimated at each time step during the peak period.

### 3.1.4 Rainfall spatial scale based on fractional coverage of basin by storm core

In this work, a different approach to classify rainfall events is presented, considering storm spatial and temporal variability in combination with rainfall intensity thresholds. To select the thresholds $Z$ for the 9 rainfall events over the radar grid (6 km x 6 km), percentiles at 25%, 50%, 75% and 95% of the entire 100 m - 1 min resolution rainfall dataset were calculated. In this way it was possible to calculate the different thresholds $Z_{25}, Z_{50}, Z_{75}$ and $Z_{95}$, corresponding to the 25%-,50%-, 75%- and 95%-ile.

Fractional coverage was largely studied in the literature and it was shown that it has a strong influence on flood response (Syed et al., 2003; ten Veldhuis and Schleiss, 2017). The percentage of coverage $\%cov$, used in this study, was defined as the sum of the number of pixel $N_t$ above a threshold at each time step $t$ divided over the total number of pixels of the catchment $N_{tot}$ and over the total number of time steps $d$ of the event:

$$\%cov = \frac{\sum_t N_t}{N_{tot} * d}. \tag{5}$$

The percentage of coverage was calculated for each event, in order to give a first classification of the spatial rainfall variability.

### 3.1.5 Rainfall cluster classification

Since variograms provide a strongly smoothed measure of rainfall field, we used alternative metric to characterize space and time scale of storm events based on cluster identification. To analyse the spatial variability of the storm core, we identified for each rainfall event the main rainfall cluster dimension.

For each time step, the area covered by rainfall above a certain threshold was considered. Main clusters were defined as the union of rainfall pixels above a given threshold. To identify the clusters, an algorithm based on Cristiano and Gaitan (2017), has been used. The algorithm executes the following rules:

– All pixels above a certain threshold are considered.





- A pixel is included in the cluster if at least one of its boundaries borders the cluster.

- Small clusters, with an area smaller than 9 ha (about 1% of catchment area) are ignored.

- In case of more than one cluster, the average of cluster areas is considered, in order to compare the cluster size at different time steps. This happens in only few cases.

To obtain a characteristic number for each storm, cluster sizes per time step were averaged over the entire duration of rainfall event. Figure 2 presents an example of rainfall coverage at a time step $t$. Rainfall was divided considering different thresholds and the red line highlights the cluster for $Z_{75}$ in Fig. 2 (a) and for $Z_{95}$ in Fig. 2 (b). The clusters identified with yellow circles are ignored because they are too small to give a considerable contribution. In case there is more than one cluster, as for Fig. 2 (b), the average of the main clusters is considered.

### 3.1.6 Maximum wetness period above rainfall threshold

To identify characteristic time scale of rainfall events, maximum wetness periods were defined as the number of time steps was estimated for which rainfall at a pixel is constantly above a given threshold. With this aim, every pixel in the catchment was analysed and maximum number of consecutive time steps above the chosen threshold was retrieved. Figure 2 (c) illustrates of the process followed to select the maximum duration $Tw_{max}$ above the threshold $Z$. For each pixel, the value of the maximum

duration above the threshold is identified. These values are averaged over the whole catchment to obtain a temporal length scale that characterizes rainfall event $Tw_Z$.

For each pixel $n$, the maximum wetness period $Tw_Z$ above a selected threshold $Z$ is defined as $\dfrac{\sum\limits_{n}^{N} Tw_{max}}{\sum N}$, where $N$ is the total number of pixels.

In order to characterize the intermittency of rainfall events, the maximum dry period $Td_{max}$, defined as the maximum

number of time steps during which the threshold $Z$ was not exceeded, was also identified. Figure 2 (c) shows how these lengths, $Tw_Z$ and $Td_Z$, were selected. The combination of these two parameters gives an indication of how constant or intermittent is the rainfall event.

### 3.2 Characterizing hydrological models' spatial and temporal scales

### 3.2.1 Models' spatial scales

Several studies have shown that drainage area is one of the dominating factors affecting the variation in urban hydrological responses resulting from using rainfall at different spatial and temporal resolutions as input (Berne et al., 2004; Ochoa-Rodriguez et al., 2015; Yang et al., 2016). Considering a bigger drainage area implies aggregating and averaging rainfall and consequently smoothing rainfall peaks, with the result of having big areas that are less sensitive to high resolution measurements.

In order to compare spatial scale of models and rainfall spatial variability, the average dimension of subachatchments was

analysed to characterize the model spatial scales. To investigate the effects of the drainage area $A_d$ on hydrological response





sensitivity, thirteen locations, with connected surface that varies from less than 1 ha to more than 600 ha, were considered. Given that the coarser resolution model (SD1) does not contain small drainage areas (<35ha), only eight of the thirteen selected locations were available for SD1. To compare FD with SD models, we assumed that FD sub-catchments have the same dimension of SD2 sub-catchments. Table 1(b) presents the drainage area $A_d$ connected to each location, while in Figure 1 the location of the selected pipes is highlighted on the catchment, with a red thick line.

Dimensionless parameters as proposed by Bruni et al. (2015) and Ogden and Julien (1994) were determined to investigate the interaction and relation between rainfall resolution and different model properties and characteristic. The *catchment sampling number* $\frac{\Delta s}{L_C}$ was introduced as the ratio between the rainfall spatial resolution $\Delta s$ and the characteristic length of the catchment $L_C$ (square root of the total area). This parameter describes the interaction between rainfall resolution and study area. If the catchment sampling number is higher than 1, rainfall variability is insufficiently captured and for small rainfall events the position might not be properly represented. The *runoff sampling number* was defined as $\frac{\Delta s}{L_{RA}}$, where $L_{RA}$ indicates the spatial resolution of the runoff model, defined as the square root of the averaged sub-catchment size (Bruni et al., 2015). Lower values of this ratio indicate that the model is unable to capture rainfall variability, while higher values indicate possible incorrect transformation of rainfall into runoff. The *sewer sampling number* $\frac{\Delta s}{L_S}$, describes the interaction between rainfall resolution and sewer length $L_S$, indicating higher sensitivity to rainfall variability with increasing values of this ratio.

### 3.2.2 Models' temporal scales

In the literature, there is no unique parameter to characterize the temporal variability of the model. Several authors have proposed different time scale characteristics (see Cristiano et al. (2017) for a review), but no unique formulation has been chosen yet, especially for urban areas. Time of concentration (McCuen et al., 1984; Singh, 1997; Musy and Higy, 2010) and lag time (Berne et al., 2004; Marchi et al., 2010) are the most commonly used temporal model scales, but other time length have been proposed in the literature (Ogden et al., 1995; Morin et al., 2001). In this study, temporal variability of the three models was classified using lag time $t_{lag}$, which describes the runoff delay compared to rainfall input. $t_{lag}$ can be defined in different ways: as difference between the centroid of hyetograph and the centroid of hydrograph (Berne et al., 2004), or as the distance between rainfall and flow peaks (Marchi et al., 2010; Yao et al., 2016). Hyetograph in a specific location was estimated as average of rainfall intensity that interests the considered sub-catchment, while the hydrograph was represented using the flow in selected pipes. The lag time can be considered as a characteristic basin element. It depends on drainage area size, slope and imperviousness (Gericke and Smithers, 2014; Morin et al., 2001; Berne et al., 2004; Yao et al., 2016), but it is also influenced by rainfall characteristics. For this reason, $t_{lag}$ was calculated for the nine rainfall events and the average of these values was taken as representative number.

Lag time increases with drainage area, following a power law as proposed by Berne et al. (2004). For urban areas, an empirical relation between catchment area $A$ (ha) and lag time $t_{lag}$ (min) was presented:

$$t_{lag} = 3A^{0.3}. \tag{6}$$



This relation was confirmed, incorporating results obtained by Schaake and Knapp (1967) and Morin et al. (2001). $t_{lag}$ was calculated for each selected sub-catchments, and then compared with the rainfall temporal scale, to investigate the interaction between model and rainfall scale. The relation between averaged lag time and connected drainage area was studied at each location.

**3.3 Statistical indicator for analysing rainfall sensitivity**

To investigate effects of rainfall aggregation on peak intensity, the peak attenuation ratio $Re_R$ was calculated for rainfall. This parameter represents peak underestimation, when aggregating in space and time and it was defined as:

$$Re_R = \frac{P_{st} - P_{ref}}{P_{st}} \qquad (7)$$

where $P_{ref}$ is the peak of the measured rainfall at 100 m - 1 min resolution and $P_{st}$ is the rainfall peak at the aggregated resolution $s$ in space and $t$ in time. $P$ values vary from 0 to 1, condition for which there is no underestimation.

The coefficient of determination $R_R^2$ was used to describe rainfall intensity sensitivity to aggregation in space and time. $R_R^2$ represents the portion of variance of dependent variable that is predictable from the independent one. This parameter indicates how well regression approximates real data points. $R_R^2$ values can varies between 1 and 0, where 1 represents the perfect match between observed values $R_{ref}$ and aggregated one $R_{st}$ at spatial resolution $s$ and temporal resolution $t$.

**3.4 Statistical indicators for analysing hydrological response**

Rainfall was synthetically applied over models and flow and depth were calculated in 13 selected locations, to study the hydrological response and to compare the three models. Following Ochoa-Rodriguez et al. (2015), rainfall was applied in such way that the storm main direction was parallel to the main direction of flow in pipes and the rainfall grid centroid coincided with the catchment centroid.

Using aggregated rainfall data as input and hydrodynamic simulation results derived from the highest-resolution rainfall (100 m and 1 min) as reference, the following two statistical indicators were calculated and analysed to quantify the influence of rainfall input resolution, at selected locations.

- Relative Error in peak flow $Re_Q$

  $Re_{Qst} = \frac{Qmax_{st} - Qmax_{ref}}{Qmax_{ref}}$ where $Re_{st}$ is the relative error in peak ($Qmax_{st}$) corresponding to a rainfall input of spatial resolution $s$ and temporal resolution $t$, in relation to the reference (100 m - 1 min) flow peak, $Qmax_{ref}$ (Ochoa-Rodriguez et al., 2015). $Re_{st}$ values bigger than zero indicate an overestimation of the peak associated to the rainfall input $st$, and vice versa, $Re_{st}$ values smaller than zero indicate an underestimation.

- Coefficient of determination $R_Q^2$

  $R_Q^2$, as described in Section 3.4 for rainfall, was applied also to the flow, to investigate effects of rainfall aggregation on hydrological response.





### 3.5 Scaling factors characterising rainfall and model scales

To investigate the impact of spatial and temporal scales of rainfall events on the sensitivity of simulated runoff to different rainfall input resolutions, Ochoa-Rodriguez et al. (2015) defined spatial and temporal scaling factors, $\theta_S$ and $\theta_T$. These factors were defined as the ratio between required spatial and temporal minimum resolutions, $\Delta s_r$ and $\Delta t_r$, and spatial and temporal

resolutions considered as input $\Delta s$ and $\Delta t$: $\theta_S = \frac{\Delta s_r}{\Delta s}$ and $\theta_T = \frac{\Delta t_r}{\Delta t}$. The combined effects of spatial and temporal characteristics were evaluated defining a combined spatial−temporal factor which accounts for spatial−temporal scaling anisotropy factor $H_t$ (Ochoa-Rodriguez et al., 2015). The anisotropy factor represents the relation between spatial and temporal scales, assuming that atmospheric properties and Kolgomorov?s theory (Kolgomorov, 1962) are valid also for rainfall (Marsan et al., 1996; Deidda, 2000; Gires et al., 2011). Combined spatial-temporal factor is then defined as: $\theta_{ST} = \theta_S * \theta_T^{\frac{1}{1-H_t}}$, where $H_t$

usually assumes the value of $1/3$ (Marsan et al., 1996; Gires et al., 2011, 2012).

Building on the work of Ochoa-Rodriguez et al. (2015), we proposed spatial and temporal scaling rainfall factors, $\delta_S$ and $\delta_T$. Rainfall cluster classification and maximum wetness period were used to describe the rainfall scale. The 75%-ile threshold was chosen as reference, accordingly to the results presented in Section4.4.3. The rainfall factors are defined as ratio between cluster dimension $S_{Z75}$ above $Z_{75}$ and maximum wetness period $Tw_{Z75}$ above $Z_{75}$ and spatial and temporal rainfall resolutions:

$$\delta_S = \frac{\sqrt{S_{Z75}}}{\Delta s} \qquad\qquad \delta_T = \frac{Tw_{Z75}}{\Delta t} \qquad\qquad (8)$$

The characteristic spatial length of the main cluster, corresponding to the square root of the main cluster, was used to define the spatial rainfall scaling factor. Combined effects of spatial and temporal rainfall scale were investigated defining $\delta_{ST}$ as a combination of $\delta_S$ and $\delta_T$.

$$\delta_{ST} = \delta_S * \delta_T \qquad\qquad (9)$$

The coefficient of anisotropy was not considered for the new parameters. The assumption that the anisotropy observed in the atmosphere is present also in the hydrological response is not always applicable. Results were however investigated with and without the anisotropy and no big differences were identified.

A similar concept was applied to model characteristics and spatial and temporal model scaling factors were defined. These factors were obtained comparing model characteristic length (square root of drainage area $A_d$) and lag time $t_{lag}$ with spatial

and temporal resolution relatively.

$$\gamma_S = \frac{\sqrt{A_d}}{\Delta s} \qquad\qquad \gamma_T = \frac{t_{lag}}{\Delta t} \qquad\qquad (10)$$

The combined model scaling factor was defined as:

$$\gamma_{ST} = \gamma_S * \gamma_T \qquad\qquad (11)$$

With the aim to identify a factor that represents the behaviour of hydrological response sensitivity well, three new parameters

are presented. The first factor is $\alpha_1$, which accounts only for the spatial aspects of model and rainfall variability. $\alpha_1$ was defined





as:

$$\alpha_1 = \frac{\sqrt{S_{Z75} * A_d}}{\Delta s^2} \tag{12}$$

A second possible way to combine rainfall and model characteristics was $\alpha_2$:

$$\alpha_2 = \frac{\sqrt{S_{Z75}}}{\Delta s} * \frac{t_{lag}}{\Delta t} = \delta_S * \gamma_T \tag{13}$$

In this case, both spatial and temporal aspects were considered. The catchment temporal scaling factor represents both spatial and temporal variability of the catchment, because of the strong relationship between lag time and drainage area described in Section 3.2.2.

The third scaling factor, $\alpha_3$, combines all spatial and temporal rainfall and model characteristics. $\alpha_3$ was defined as:

$$\alpha_3 = \frac{\sqrt{S_{Z75} * A_d}}{\Delta s^2} * \frac{Tw_{Z75} * t_{lag}}{\Delta t^2} = \delta_{ST} * \gamma_{ST} \tag{14}$$

These parameters allow to choose the best rainfall resolution or model scale to use. Depending on the available data and on the level of performance that we want to achieve, it is possible to identify the required rainfall resolution.

# 4 Results and discussion

## 4.1 Rainfall analysis

In this section, methods for quantifying rainfall space and time scales proposed in the literature (Ochoa-Rodriguez et al., 2015; Lobligeois et al., 2014), are compared to the cluster classification we propose in this paper. Additionally, change in rainfall characteristics with spatial and temporal aggregation scale will be analysed.

### 4.1.1 Spatial and temporal classification results

Spatial variability index values for each of the 9 rainfall events are presented in Table 3 for the observed rainfall at 100 m - 1 min ($I_\sigma$) and at 1000 m - 5 min ($I_{\sigma 1000m}$). Last values were added to have a direct comparison with the values presented by Lobligeois et al. (2014), using the same resolution. $I_\sigma$ values are generally high when compared to values found by Lobligeois et al. (2014) for all the investigated regions. This indicates that most events are characterised by high spatial variability. Aggregation has a strong impact on this parameter, which becomes smaller with a coarser resolution, highlighting the fact that information about rainfall variability is lost during the coarsening process. $I_{\sigma 1000m}$ values are generally higher than values presented for the Northern region, where values are below 1, but are comparable to the Mediterranean area, where $I_\sigma$ reaches values around 4.

Values obtained based on variogram analysis (spatial range) and storm tracking (temporal development) following Ochoa-Rodriguez et al. (2015) are also presented in Table 3.

Results show that the spatial variability index tends to increase as well as the the required spatial resolution for storms larger than 2500 m spatial range, while events with small spatial range (E5, E7 and E9, spatial range below 2500 m) are characterised





by relatively high spatial variability indexes. Required temporal resolution $\Delta t_r$, obtained from the combination of velocity and required spatial resolution (see Section 3.1.3) varies between 1.7 and 5.9 minutes; lowest values of $\Delta t_r$ are associated with fast storm events (e.g. E8 and E5) and small-scale events (e.g. E9 and E7).

### 4.1.2 Thresholds and percentage of coverage

The first step in obtaining cluster dimensions is to identify rainfall thresholds ($Z$) characterising the rainfall values' distribution (see Section 3.1.4). Table 4 shows rainfall threshold values corresponding to the 25-, 50-, 75- and 90-%iles for the 9 rainfall events.

The 25%-ile of the rainfall values distribution is zero, indicative of strong intermittency and small areal coverage of some of the events (especially events E7 and E9). The 95%-ile is 22 mm/h (over a 1-minute time window), corresponding to a
recurrence interval of less than a half year (KNMI, 2011), indicating that the selected events are representative of frequently occurring events. For this region, rainfall intensities above 25 mm/h, over a 15-minute time window, correspond to a return period of once per year, indicating an intense rainfall event. For only few rainfall events, E1, E2, E3 and E7, the 25 mm/h threshold is exceeded over a 15-minute time window, for few time steps and, in particular, for E7 this happens only at the peak. This implies that rainfall events considered in this study are not classifiable as extreme.

The percentage of areal coverage, estimated for the catchment, is presented in Fig. 3(a, d, g, j). Areal coverage associated with 25%-ile values provides an indication of event scale intermittency. Events with 25%-iles close to 1 cover the entire catchment most of the time, while smaller and more intermittent events, especially E7 and E9, are characterised by lower 25%-ile values. Areal coverage for 95%-ile thresholds indicates the size of storm cell cores: E1 and E2 have storm cores covering up to 65-70% of the catchment; E4 and E6 have median coverage values close to zero, indicating that these are mild events without an intense
storm core.

Boxplots in Fig. 3 (b, e, h, k) show the number of time steps above selected thresholds as a percentage of total event duration, to enable comparison between events. Results confirm patterns identified based on areal coverage: events E7 and E9 are identified as high intermittency events (based on 25%-ile threshold). Maximum percentage of time steps above the highest threshold is 30% for events E1 and E2. Each boxplot represents the spatial variability of rainfall between pixels. Thresholds $Z_{50}$
and $Z_{75}$ present a high intra-event variability, highlighting the differences between rainfall events. For the other two thresholds, the intra-event variability is not high, suggesting that the rainfall event characteristics might not be well represented. For $Z_{95}$, all events present a coverage variability lower than 30%, and differences between events are not properly defined. Thresholds $Z_{50}$ and $Z_{75}$ present also a high inter-event variability, indicating that in these cases the spatial variability of the rainfall event above the catchment area is high.

### 4.1.3 Rainfall cluster classification

Dimensions of the main cluster were determined for each of the four thresholds and for all time steps of the nine events. Results are presented in Fig. 3 (c, f, i, l), where the red line indicates the median and the blue dot the average.





The plots show that for $Z_{25}$ only intermittent events, like E7 and E9, present a median below 861 ha (entire catchment area). The intra-event variability is generally quite high for most of the events, especially for the 50%-ile and 75%-ile, indicating that clusters change their dimension and shape during the event. Only few events, E4 and E2, do not show high variability above $Z_{25}$ and $Z_{50}$ threshold. For $Z_{95}$, the cluster dimension variability is relatively small, suggesting that the average or the median

can be a good approximation of the storm core dimension. Values above $Z_{50}$ present high inter-event variability. There is a clear distinction between constant events, such as E2 and E4, and intermittent events, E7 and E9, which show low median and average values.

Intense and constant rainfall events are also characterized by median values being generally higher than the mean. On the other hand, intermittent events, such as E9, have an average higher than the median, especially for the 50%- and 75%-ile. This

results suggest that $Z_{50}$ and $Z_{75}$ are able to describe well rainfall spatial and temporal scale.

### 4.1.4   Maximum wet and dry period

The maximum wet period $Tw_Z$ and maximum dry period $Td_Z$ were calculated for four rainfall intensity thresholds in order to represent temporal variability of a rainfall event. Table 5 presents maximum wetness period $Tw_Z$ and maximum dry period $Td_Z$, normalized by total duration of the rainfall event, to enable comparison between events and to investigate how long the

main core is in relation to the total duration of the event.

For some events $Tw_Z$ decreases depending on the threshold, passing from values close to 1 for $Z_{25}$ to values close to 0 for $Z_{95}$. The change between different thresholds can be gradual, as for example for E2, E8 or E5, or sharp, as is the case of E3 or E4. For intermittent events, on the other hand, the maximum wet period does not vary too much, and it is relatively short, like E7 or E9. This implies that there are probably multiple short periods above the threshold. When comparing $Tw_Z$ and $Td_Z$,

we can observe that some events show a symmetrical behaviour, when decrease in wet period coincides with increase in dry period, with the increase of the threshold (E4, E3). E7 and E9 present a moderate decrease of $Tw_Z$ while they have a steep increase of $Td_Z$, indicative of strong intermittency. For the other events, the behaviour is generally the opposite, indicative of a concentrate storm core.

### 4.2   Hydrological models, spatial and temporal scales

### 4.2.1   Spatial model scale

Dimensionless sampling numbers, presented at first by Ogden and Julien (1994), and then re-proposed by Bruni et al. (2015), are presented in Table 6 for the three models (for underlying equations see Section 3.2.1). SD2 and FD model have the same contributing area and network length, hence they show that values for the catchment sampling number and sewer sampling number are the same.

Catchment sampling numbers higher than 1 indicate that models can not properly represent rainfall variability (Bruni et al., 2015). In this study, for 3000 m spatial rainfall resolution values are bigger than 1, so poor model performance at this resolution is expected. The runoff sampling number suggests that SD1 will not be able to capture rainfall variability, because it presents





low values for all spatial resolutions, while FD has high values of this parameter, which highlights some uncertainty in rainfall-runoff transformation. SD2, instead, presents runoff sampling numbers similar to the values found by Bruni et al. (2015), where this parameter varied between 2.6 for high resolution and 93 for lower resolution. The sewer sampling number applied to SD2 and FD, presents similar results to Bruni et al. (2015), where the values were varying between 2 for high resolution and 77

for low resolution. On the other hand, the sewer sampling number is pretty low for SD1, which indicates a low sensitivity of this model to rainfall variability. This parameter increases with coarsening of spatial resolution, suggesting a high sensitivity to coarser rainfall resolutions.

The catchment sampling number can be applied also to the selected sub-catchments, comparing spatial resolution with the sub-catchments dimension reported in Table1(b). Also in this case, when the ratio is bigger than 1 the rainfall might not be

well represented. This happens for sub-catchment L1, which is smaller than 100 m, and for all locations when they have to deal with 3000 m rainfall resolution. Locations from L2 to L5, presenting a drainage area between 100 m and 500 m, should show the effects of aggregation for spatial resolution of 500 m and 1000 m, when the catchment sampling coefficient is higher than 1, and the variability is not well captured. When the catchment sampling number is lower than 0.2, the catchment is too large to be compared to the rainfall input, and the effects of averaging over the area should be visible, as for example for L13 when

considering a 100 m input resolution.

### 4.2.2   Temporal model scale

Lag time $t_{lag}$ was computed for nine storms for each model at twelve sub-catchments and at the catchment outlet, as explained in Section 3.2.2. Results, presented in Fig. 4 (a), show that $t_{lag}$ increases with drainage area and varies from just above 1 min for FD at L1 (upstream location with the smallest $A_d$) to over 100 min for the coarsest model and largest catchment scale.

Only for a few locations, $t_{lag}$ is lower than 10 min and for this reason a low sensitivity to temporal variability of rainfall events is expected. On the other hand, lag times vary over a wide range between events and this highlights a strong influence of event characteristics. Model scale clearly influences computed lag times, which are generally larger for coarser model, where sub-catchments are bigger. However, for locations with smaller drainage area ($< 245$ ha), SD1 presents $t_{lag}$ values comparable with the other models, but with a much lower variability compared to the finer scale models.

As discussed in Section 3.2.2, $t_{lag}$ strongly depends on drainage area. Figure 4(b) shows how lag time varies, as a function of drainage area, for SD2, based on average, median, minimum and maximum values across rainfall events. Results confirm that $t_{lag}$ increases with the drainage area, fitting a power law, similar to the one suggested by Berne et al. (2004) (eq. 6). In this case the power law that fits at best the average of empirical data is $t_{lag} = 8.9 * A_d^{0.27}$ ($R^2 = 0.841$), equation that presents the same exponent of the one proposed by Berne et al. (2004) and slightly higher coefficient. The power law proposed by Berne

et al. (2004) represents a wider range of surface areas wider than what is presented in this work, hence only a small part of it is considered.





### 4.3 Sensitivity of rainfall: effects of spatial and temporal aggregation on rainfall peak and distribution

#### 4.3.1 Effects of aggregating on the maximum rainfall intensity at catchment scale

Figure 5 presents rainfall peak attenuation ratios $Re_R$ for the range of spatial and temporal aggregation levels investigated. The plot shows the median over the nine events (marker) and the variability of the data (from 25% to 75% solid lines and total range dotted lines).

Rainfall peaks are reduced up to 80% when aggregating in space or time and up to 88% when combining the spatial and temporal aggregation at the coarsest resolution. For high resolution, aggregation over time seems to play a larger role then over space. Aggregating from 1 min to 3 min approximately half of the rainfall peak is lost, while from 100 m to 500 m peak attenuation is relatively smaller (40%). For lower resolutions, spatial aggregation has a slightly stronger attenuating effect than temporal aggregation. At 3000 m spatial resolution, rainfall peaks are strongly underestimated, independent of the temporal resolution.

#### 4.3.2 Rainfall aggregation analysis at sub-catchment scale

In this sub-section, we compare effects of spatial and temporal aggregation on rainfall variability and peak intensity across sub-catchment scales. Figure 6 shows examples of rainfall aggregation effects, as a function of the drainage area. Results for two rainfall events are shown: E4 is a constant, low intensity event, which has a low variability in time and space, while E9 is an intermittent events, with multiple peaks. The plots clearly show that rainfall variability for the constant event is less sensitive to aggregation than that for the intermittent event. Rainfall sensitivity to aggregation decreases for larger size. $Re_R$ and $R_R^2$ results for all the 9 studied events are available in the supplement.

### 4.4 Rainfall and model influence on hydrological response

#### 4.4.1 Sensitivity of the hydrological response to rainfall input resolution

Figure 7 shows results for statistical indicators $Re_Q$ and $R_Q^2$ for sixteen combinations of rainfall resolution and in relation to catchment area. Results are shown for a stratiform low intensity rainfall event (E4) and a convective intermittent storm (E9)for increasing catchment. For both events, the sensitivity to rainfall input resolution generally decreases for increasing of catchment size. Variability of $Re_Q$ and $R_Q^2$ is much stronger for E9 than E4, pointing out the important role of rain event characteristics.

Comparing Fig. 6 with Fig. 7, similar patterns are observed for rainfall and flow. In both cases, sensitivity to rainfall aggregation in space and time decreases with increase of the drainage area. Moreover, in both cases, the small and constant event (E4) is less sensitive to aggregation than the intermittent one (E9). Rainfall patterns are more sensitive to aggregation than flow, due to smoothing induced by rainfall runoff processes.





### 4.4.2 Influence of the model complexity on hydrological response sensitivity

To investigate the influence that model complexity has on hydrological response sensitivity, results obtained with the three models are analysed. Figure 8 compares the influence of model complexity to the impact of spatial rainfall variability on the sensitivity of hydrological response. For each model, outputs at all locations are plotted for the 16 different rainfall input resolutions. There is not a clear behaviour that characterizes differences between sensitivity of the three models. All models appear sensitive to 3000 m spatial resolution and 10 min temporal resolution: in these cases the performance is lower. For upstream location, SD1 seems to be slightly more sensitive than the other models to spatial coarsening for the upstream location, while FD performs worse for L13. The plot shows that there are some minor differences between the outputs of the three models, but the strongest sensitivity is connected to the rainfall scale as characterized by the cluster dimension. All models show higher sensitivity to small clusters, especially for cluster sizes below 100 ha. For small clusters, SD1 presents a higher sensitivity for both statistical indicators, while it is less sensitive than SD2 and FD for large clusters.

Model complexity does not have a large influence on sensitivity to rainfall resolution coarsening, while other characteristics, such as rainfall parameters or catchment details, seem to have a higher impact.

### 4.4.3 Influence of rainfall scale classification on hydrological response

Several approaches to classify rainfall variability have been presented and discussed in Section 3.1 and in Section 4.1. In these sections, their influence on the hydrological response will be analysed.

Figure 9 compares the influence of spatial and temporal required resolutions ($\Delta s_r$ and $\Delta t_r$), spatial variability index $I_\sigma$, cluster above $Z_{75}$ and $Z_{95}$ and the maximum wet period $Tw_{Z75}$ to model performance for different resolutions. Sensitivity to rainfall input resolution generally increases for smaller required spatial and temporal resolution, for higher spatial variability index and for smaller cluster size. Clearest relationships are observed for required temporal resolution and cluster size above $Z_{75}$. This parameter seems to represent quite well spatial scale of the rainfall events, and therefore is chosen in this work to characterize the spatial scale of rainfall events.

Figure 10 compares the influence of rainfall spatial scale, based on cluster size above $Z_{75}$, with drainage area size. Variability of $R_Q^2$ is higher for lower values of both rainfall scale and drainage area and decreases in a similar way with increase in both rainfall and catchment dimensions.

For this study case, we can conclude that sensitivity to rainfall resolution depends mainly on the scale of rainfall events and study catchment, and much less on complexity of models used. Choosing a complex model is useful only when studying small scale events and catchments only if high resolution rainfall data are available.

### 4.5 Rainfall and model scaling factors

Spatial, temporal and combined scaling factors proposed by Ochoa-Rodriguez et al. (2015) were calculated for this study and are presented in Fig. 11 (a-c). Higher values of the scaling factors $\theta_S$, $\theta_T$ and $\theta_{ST}$ are generally associated with higher





modelling performance, expressed in terms of $R^2$. The combined spatial-temporal scaling factor, $\theta_{ST}$, in particular indicates how high $R_Q^2$ values are obtained for $\theta_{ST} > 15$ ($R^2 > 0.9$).

As discussed in Section 4.4.3, both rainfall scale and catchment characteristics strongly affect sensitivity of hydrological response to rainfall resolution. For this reason, the new dimensionless factors proposed combine rainfall and catchment prop-

erties. From results shown in Fig. 11 (a-c), spatial variability seems to have a better relation with the sensitivity variability than the temporal scale. The factor $\alpha_1$ takes into account only the spatial scale of model and rainfall variability. Figure 11(d) shows $R_Q^2$ as a function of $\alpha 1$. The plot presents a clear trend, indicating low model performance for low values of $\alpha_1$ and high performance for values of $\alpha_1$ larger than 100.

Figure 11(e) shows $\alpha_2$ and response sensitivity. For values of $\alpha_2 > 40$, $R_Q^2$ is higher than 0.95, indicating a very good

performance. For values of $\alpha_2 < 10$, $R_Q^2$ is lower than 0.8. Figure 12(b) shows the same plot on a logarithmic scale, which better visualises thresholds of performance. Different resolutions are highlighted in the plot. Low resolution in space generally lead to a lower $\alpha$'s values than low temporal resolution, and consequently to a lower performance of the model.

Figure 11(f) and Fig. 12(c) plots $R_Q^2$ against $\alpha_3$. Figure 12(c) indicates that for values of $\alpha_3$ higher than 3000 a high performance of $R_Q^2$ is guaranteed ($R^2 > 0.90$). For $400 < \alpha_3 < 3000$ the performance of $R_Q^2$ drops to 0.8.

Comparing the scaling factors, we observe that $\alpha_2$ works better in distinguishing critical resolutions for a given model performance. There are indeed, less points with high $R_Q^2$ below the identified thresholds. Moreover, $\alpha_2$ should be preferred because it allows to use fewer parameters, without losing information about temporal characteristics, as it is for $\alpha_1$.

## 5  Conclusions

In this study we investigated effects of rainfall and catchment scales on sensitivity of urban hydrological models to different

rainfall input resolutions. The aim was to identify dimensionless ratios of storm and catchment scales that support critical resolution for reproducing hydrological response. Cranbrook, a small urbanized area of 861 ha, was analysed with the help of two semi-distributed and a fully distributed models. Rainfall data measured by a dual polarimetric X-band radar at 100 m and 1 min resolution, were aggregated to obtain different rainfall resolutions and then used as input for the hydrological models. A new rainfall classification method, based on cluster identification was presented in this work. Different rainfall classification

methods were used to characterize storm event scales.

From this work we draw the following conclusions.

- Rainfall classification based on clustering is an easy and fast method to quantify spatial scale of rainfall events. In particular, rainfall clusters associated with the 75%-ile threshold turned out to give a realistic approximation of the spatial dimension of the storm core.

- Spatial and temporal aggregation of rainfall data can have a strong effect on rainfall peak and intensity. Rainfall peaks were reduced up to 80% when aggregating in space to 3000 m resolution or in time at 10 min resolution. Both space and time have a strong influence on peak attenuation. Temporal aggregation has a stronger influence at 1 - 5 min resolution, while aggregation in space has bigger impact at low (1000 - 3000 m) resolution.




- Lag time estimated for the investigated sub-catchments was used to represent the temporal characteristics of models. Lag time increased with the catchment area size, yet varied strongly between events (approx. by a factor of 2, 25-75%-ile range). Mean lag time fitted an empirical power law similar to the one proposed by Berne et al. (2004), yet with a higher intercept.

- Effects of rainfall aggregation in space and time on hydrological response depend on rainfall event characteristics. Rainfall events with constant intensity are less affected by aggregation than small scale intermittent events. However, results showed that aggregation effects are stronger for rainfall than flow. Results showed that smoothing of rainfall peak intensities by aggregation was much stronger than for flows. Rainfall aggregation effects on hydrological response are smoothed during the rainfall runoff transformation processes.

- For the case study under consideration, model spatial resolution does not appear to have a big impact on hydrological response sensitivity to rainfall input resolution. Three models of different complexity were all sensitive to rainfall resolution. Low resolution model was more sensitive to rainfall resolution for small scale storms, while the high resolution fully distributed model showed stronger sensitivity at larger catchment scale.

- Rainfall and catchment scales were shown to have a strong impact on hydrological response sensitivity. This indicates
15
that the relation between rainfall and catchment scale needs to be taken into account when investigating the hydrological response of a system.

- New spatial, temporal and combined scaling factors were introduced to analyse hydrological response sensitivity to rainfall resolution. These dimensionless scaling factors combine rainfall scale, model scale and rainfall input resolution and enable identification of critical rainfall resolution thresholds to achieve a given level of accuracy. Thus, the scaling factors
20
support selection of adequate rainfall resolution to obtain a certain level of accuracy in the calculation of hydrological response.

However, there are still some aspects that need further investigation. More and different rainfall events and different study areas should be investigated in order to test the applicability of the scaling factors and thresholds identified for other geographical and climatological conditions. In further work, cluster rainfall classification and dimensionless $\alpha$ parameters will be
25
investigated based on field observations in combination with modelling. Different scales will be considered to investigate the range of applicability of the scaling factors. Additionally, a better definition of temporal rainfall scale needs to be developed, with a parameter able to represent rainfall variability, highlighting the constant or intermittent character of rainfall events.

*Code availability.*  Cristiano and Gaitan (2017)

*Data availability.*



*Competing interests.* No competing interests are present.

*Acknowledgements.* The authors would like to thank Innovyze for providing a InfoWorks licence. This study was funded by the EU INTER-REG IVB RainGain Project.





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

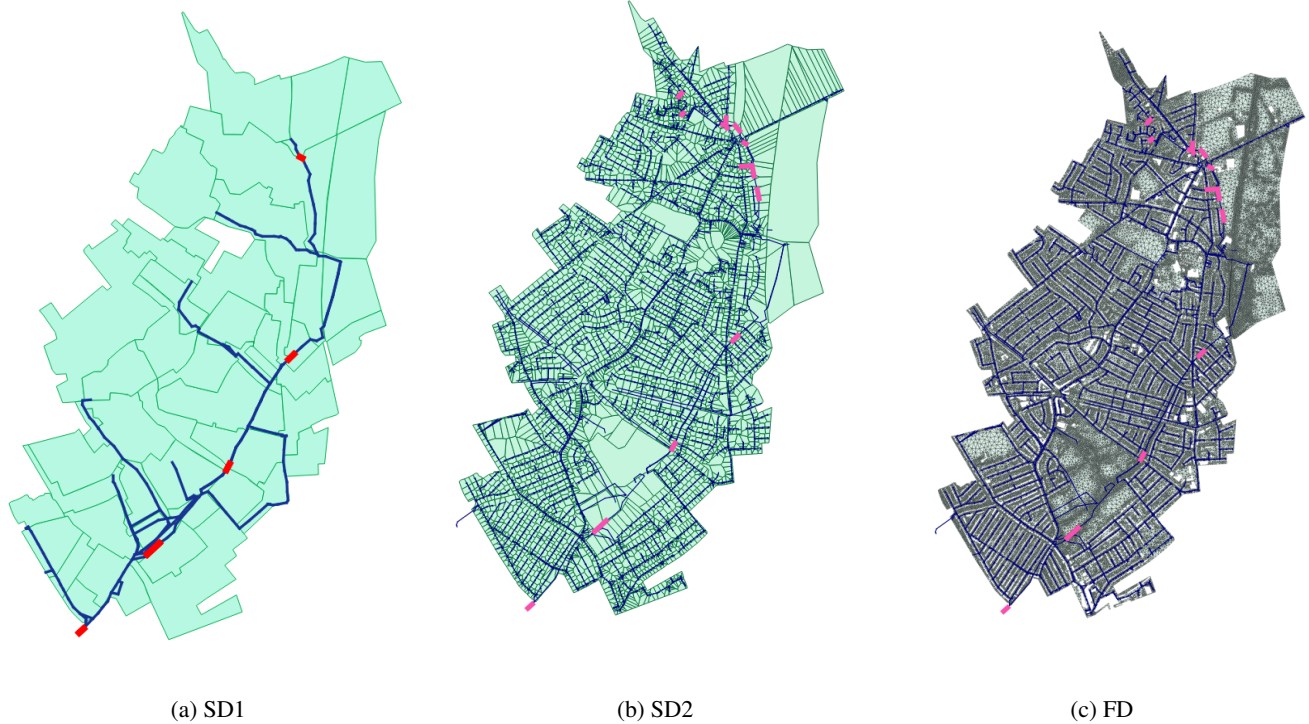

|  (a) SD1 | (b) SD2 | (c) FD |

**Figure 1.** Catchment area represented with the 3 different models: (a) SD1, (b) SD2 and (c) FD. The subdivision of the surface in sub-catchments or 2D elements is shown for each model, as well as the sewer network. The selected 13 locations/pipes are highlighted.





**Figure 2.** Illustration of rainfall cluster classification. Different colors represent different rainfall thresholds. The pixels above the same threshold are used to estimate the percentage of coverage above a certain threshold. The red line encloses the clusters above threshold $Z_{25}$ and $Z_{95}$ in (a) and (b) respectively. Single isolated pixels and small clusters (yellow dotted circles) are ignored. (c) Schematic representation of maximum wet period $Tw_Z$ (red) and the maximum dry period $Td_Z$ (light blue) for a pixel, for each threshold.



**Figure 3.** Percentage of areal coverage above selected threshold, calculated over all time steps and per rainfall event (a, d, g, j). Percentage of coverage above the selected threshold, calculated at each pixel and presented for each rainfall event(b, e, h, k). Cluster dimensions across all time steps per event for the four selected thresholds (c, f, i, l). Blue dots represent the average, green or red lines the median, boxes indicate the first to third quartile and whiskers extend 1.5 times the interquartile range below the first and above the third quartile.





(a)

(b)

**Figure 4.** Variability of the lag time, depending on the location, for each model (a). The boxplots represent the median (red line), the upper (third quartile) and lower (first quartile) quartile (boxes boundaries), and 1.5 times the interquartile range below the first and above the third quartile (whiskers). Drainage areas corresponding to each location are presented in Table 1(b). Average, median, minimum and maximum value of the lag time as function of $A_d$ for SD2.(b) Fitting power law curves and the power law relation proposed by Berne et al. (2004) are plotted.





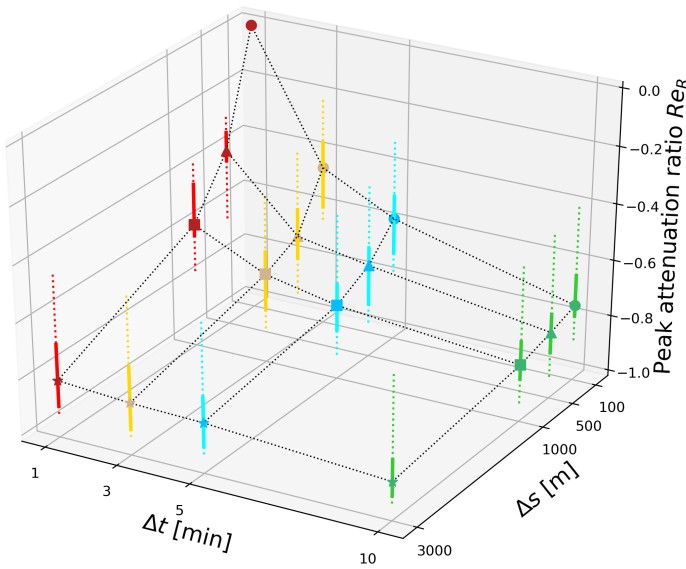

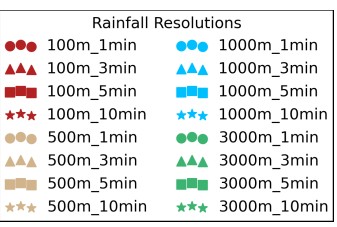

**Figure 5.** Peak attenuation ratio $Re_R$ for the 9 rainfall events, as a function od temporal and spatial rainfall resolution. Symbols indicate the median over the 9 events, solid lines represent the first to the third quartile, dotted lines vary from minimum to maximum. Colours represent different temporal resolutions and markers used for the median indicate different spatial resolutions.





**Figure 6.** Impact of aggregation in space and time on rainfall peak ($Re_R$) and overall pattern ($R_R^2$) for two selected events, as function of sub-catchment size ($A_d$). E4 is a constant low intensity event with low spatial variability. E9 is an example of intermittent event, with a high velocity. Different colors and symbols indicates different rainfall resolutions used as input. Other events are presented in the supplement material





(a)

(b)

(c)

(d)

**Figure 7.** Relative error in peak $Re_Q$ and coefficient of determination $R_Q^2$ for SD2, plotted as function of $A_d$, for the sixteen combinations of rainfall input resolutions. Two different events are presented: E4, a low-intensity constant event, and E9, a multiple peak event.



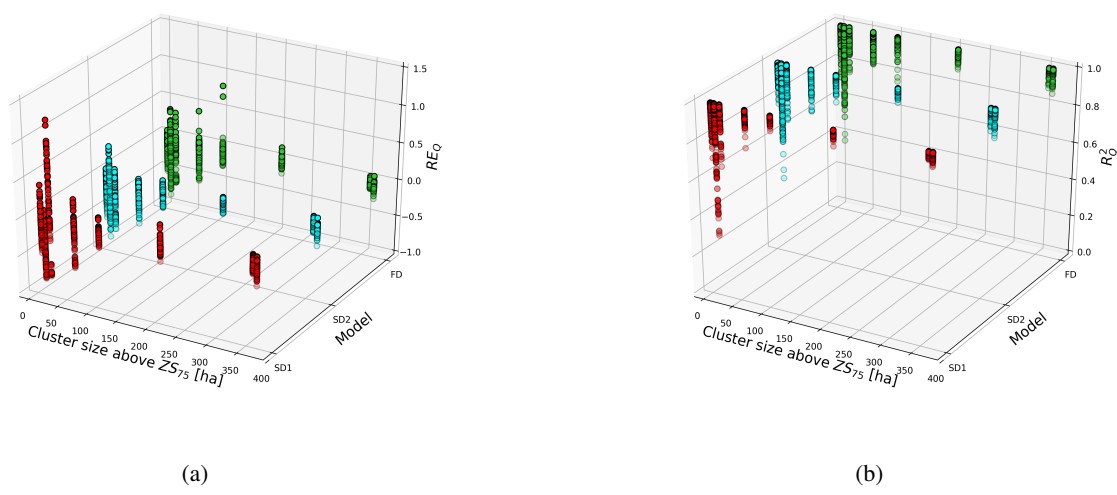

         (a)                      (b)

**Figure 8.** $Re_Q$ and $R_Q^2$ variability, in relation to model type and rainfall characterized by cluster dimension $S_{Z75}$, for all locations and all combinations of rainfall input resolution. Colours identify the three different models.



**Figure 9.** $R_Q^2$ at Loc2 for different rainfall resolution, plotted against different rainfall characterising scales: spatial (a) and temporal (b) required resolution, Spatial Variability Index (c), dimension of cluster above $Z_{75}$ (d) and $Z_{95}$ (e) and maximum wet period above $Z_{75}$ (f).





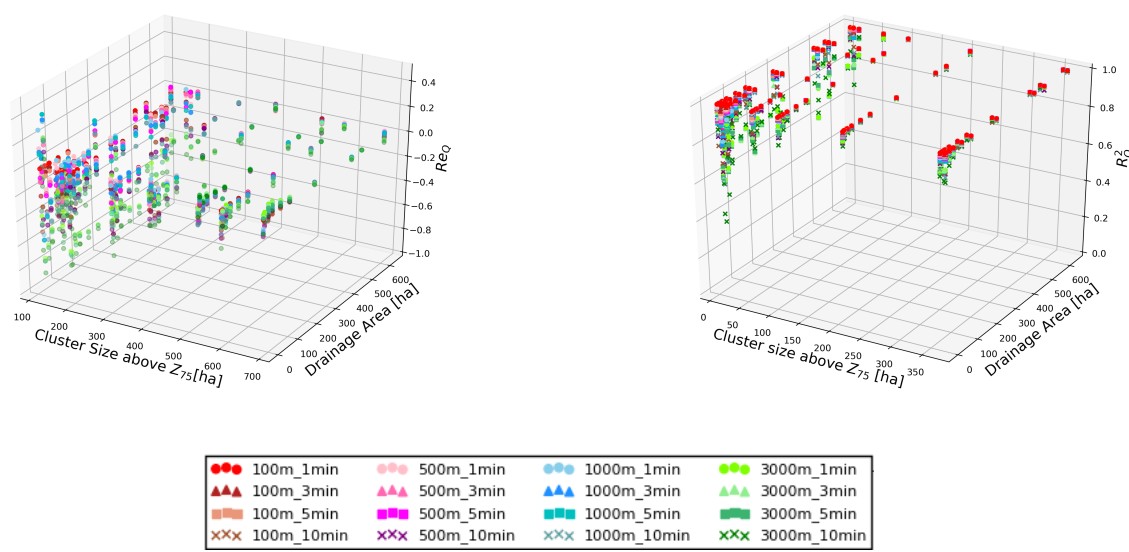

**Figure 10.** $Re_Q$ and $R_Q^2$ as function of cluster dimension above $Z_{75}$ and $A_d$. Different colors and symbols indicates different rainfall resolution input.





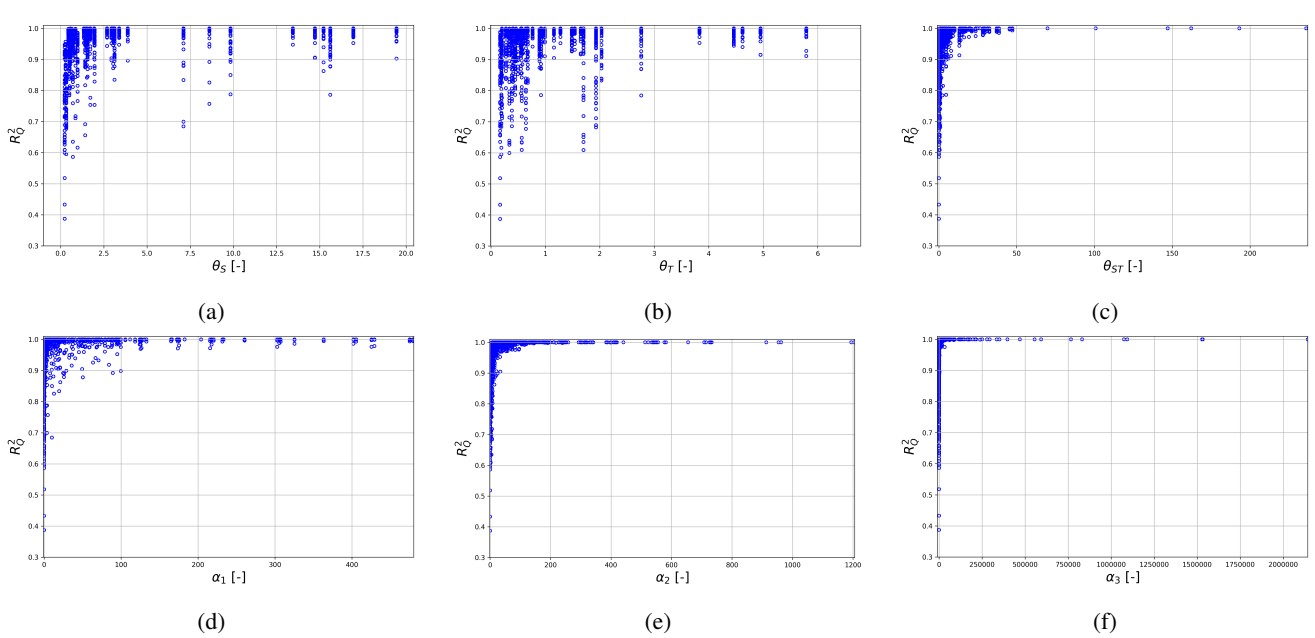

**Figure 11.** Performance statistic $R_Q^2$ as a function of dimensionless numbers $\theta_S$, $\theta_T$, $\theta_{ST}$, $\alpha_1$, $\alpha_2$, $\alpha_3$. For each parameter all events, rainfall resolutions and locations are plotted.





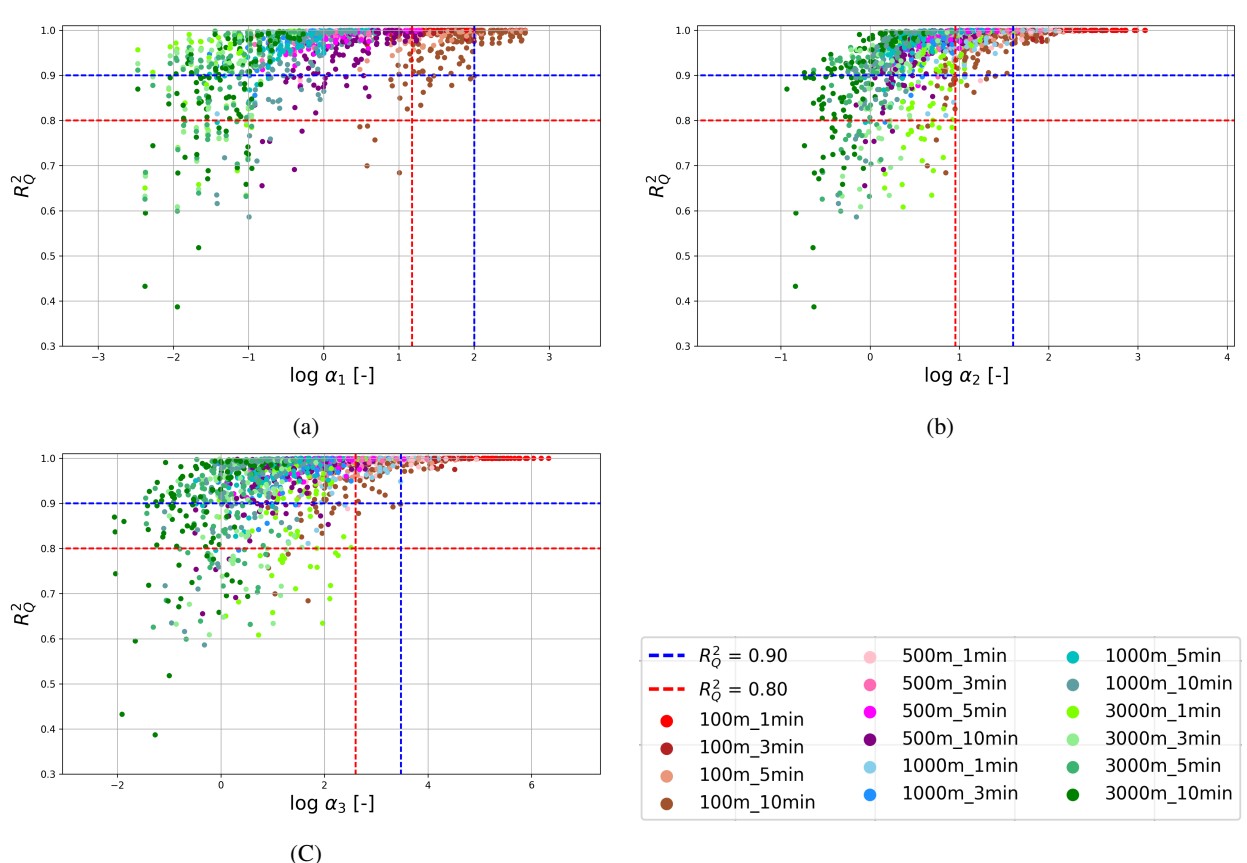

**Figure 12.** Logarithmic plots of $R_Q^2$ as function of (a) $\alpha_1$, (b) $\alpha_2$ and (c) $\alpha_3$. Different colours indicate different resolutions.





**Table 1.** (a) Summary of the hydrological model characteristics of the 3 models. (b) Drainage area connected to the investigated locations for each model

(a)

| | SD1 | SD2 | FD |
|---|---|---|---|
| # of sub-catchments | 51 | 4409 | 4367 |
| # of node | 242 | 6963 | 6963 |
| # of pipes | 270 | 6993 | 6993 |
| Catchment Area (ha) | 846 | 851 | 851 |
| Contributing % Impervious | 43 | 40 | 15 |
| Contributing % Pervious | 56 | 60 | 0 |
| Average area (ha) | 16.6 | 0.2 | 0.006* |
| St. Dev (ha) | 13.4 | 0.8 | 0.000* |
| Max (ha) | 61.8 | 40.1 | 0.099* |
| Min (ha) | 11.7 | 0.005 | 0.006* |
| Total length (km) | $\sim 16$ | $\sim 150$ | $\sim 150$ |
| N of manholes | 236 | 6207 | 6207 |
| N of 2D Elements | no | no | 117712 |

* Dimension of the 2D triangular mesh elements

(b)

| | SD1 (ha) | SD2 (ha) | FD (ha) |
|---|---|---|---|
| Loc1 | - | 0.9 | 0.9 |
| Loc2 | - | 6.7 | 6.6 |
| Loc3 | - | 9.5 | 9.5 |
| Loc4 | - | 21.3 | 21.3 |
| Loc5 | - | 24.6 | 24.6 |
| Loc6 | 36 | 42.9 | 42.9 |
| Loc7 | 80 | 43.7 | 43.7 |
| Loc8 | 80 | 83.9 | 83.9 |
| Loc9 | 137 | 129.2 | 129.2 |
| Loc10 | 290 | 254.8 | 254.8 |
| Loc11 | 484 | 448.3 | 448.3 |
| Loc12 | 538 | 502.5 | 502.5 |
| Loc13 | 846 | 626.6 | 626.6 |

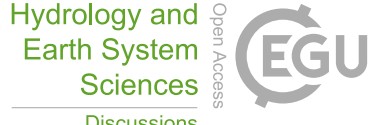



**Table 2.** Rainfall events characteristics

| Event ID | Date | Initial - ending times | Total depth (average / min / max) (mm) | Max intensity over 1 min (areal average / pixel) (mm/h) |
|---|---|---|---|---|
| E1 | 18/01/2011 | 05:10-08:00 | 31 / 18 / 46 | 32 / 1120 |
| E2 | 18/01/2011 | 05:10-08:00 | 36 / 16 / 47 | 26 / 124 |
| E3 | 28/06/2011 | 22:05-23:55 | 9 / 4 / 18 | 28 / 242 |
| E4 | 18/06/2012 | 05:55-07:10 | 10 / 8 / 12 | 12 / 24 |
| E5 | 29/10/2012 | 17:05-19:00 | 5 / 1 / 14 | 7 / 83 |
| E6 | 02/12/2012 | 00:05-03:00 | 5 / 2 / 8 | 7 / 39 |
| E7 | 23/06/2013 | 08:05-11:30 | 4 / 1 / 13 | 9 / 307 |
| E8 | 09/05/2014 | 18:15-19:35 | 4 / 1 / 9 | 13 / 67 |
| E9 | 11/05/2014 | 19:05-23:55 | 6 / 1 / 13 | 11 / 247 |





**Table 3.** Rainfall spatial and temporal characterization proposed by Ochoa-Rodriguez et al. (2015) and rainfall spatial variability index proposed by Lobligeois et al. (2014)

| Event ID | Ochoa-Rodriguez et al. (2015) | | | | Lobligeois et al. (2014) | |
| --- | --- | --- | --- | --- | --- | --- |
| | Spatial range | Mean velocity | Required spatial resolution | Required temporal r resolution | Spatial Variability Index at 100 m − 1 min | Spatial Variability Index at 1000 m − 5 min |
| | $(r)$ | $(|\bar{v}|)$ | $\Delta s_r$ | $\Delta t_r$ | $I_\sigma$ | $I_{\sigma 1000m}$ |
| | (m) | (m/s) | (m) | (min) | (-) | (-) |
| E1 | 4057 | 9.8 | 1695 | 5.8 | 12.7 | 6.4 |
| E2 | 3525 | 9.9 | 1473 | 5.0 | 7.4 | 5.2 |
| E3 | 4655 | 14.0 | 1945 | 4.6 | 10.4 | 6.5 |
| E4 | 3219 | 11.7 | 1345 | 3.8 | 2.6 | 1.5 |
| E5 | 2062 | 14.1 | 861 | 2.0 | 7.7 | 4.2 |
| E6 | 3738 | 11.7 | 1561 | 4.5 | 3.7 | 2.0 |
| E7 | 1703 | 14.0 | 711 | 1.7 | 16.6 | 5.9 |
| E8 | 3644 | 18.4 | 1523 | 2.8 | 7.9 | 4.2 |
| E9 | 2355 | 17.0 | 984 | 1.9 | 15.3 | 6.5 |





**Table 4.** Thresholds values obtained for the 9 rainfall events considered.

| Threshold | $Z_{25}$ | $Z_{50}$ | $Z_{75}$ | $Z_{95}$ |
|---|---|---|---|---|
| Percentile | 25% | 50% | 75% | 95% |
| Values | 0 mm/h | 0.5 mm/h | 7 mm/h | 22 mm/h |



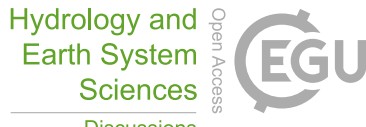

**Table 5.** Maximum wetness periods above the threshold, calculated for each pixel, averaged over the total catchment, and then divided by the total duration.

| Event ID | Maximum wet period | | | | Maximum dry period | | | |
|---|---|---|---|---|---|---|---|---|
| | $Tw_{Z25}$ | $Tw_{Z50}$ | $Tw_{Z75}$ | $Tw_{Z95}$ | $Td_{Z25}$ | $Td_{Z50}$ | $Td_{Z75}$ | $Td_{Z95}$ |
| | [-] | [-] | [-] | [-] | [-] | [-] | [-] | [-] |
| E1 | 0.53 | 0.50 | 0.42 | 0.17 | 0.16 | 0.25 | 0.27 | 0.35 |
| E2 | 0.98 | 0.74 | 0.30 | 0.06 | 0.02 | 0.07 | 0.13 | 0.30 |
| E3 | 0.97 | 0.43 | 0.10 | 0.06 | 0.02 | 0.08 | 0.63 | 0.72 |
| E4 | 1.00 | 0.98 | 0.32 | 0.01 | 0.01 | 0.02 | 0.11 | 1.00 |
| E5 | 0.77 | 0.57 | 0.14 | 0.11 | 0.11 | 0.28 | 0.38 | 0.57 |
| E6 | 0.52 | 0.24 | 0.13 | 0.12 | 0.12 | 0.29 | 0.52 | 0.99 |
| E7 | 0.28 | 0.14 | 0.13 | 0.12 | 0.13 | 0.28 | 0.53 | 0.71 |
| E8 | 0.83 | 0.43 | 0.14 | 0.07 | 0.07 | 0.22 | 0.34 | 0.53 |
| E9 | 0.22 | 0.19 | 0.18 | 0.17 | 0.17 | 0.30 | 0.56 | 0.69 |





**Table 6.** Dimensionless parameters for the three models used in this study, based on Bruni et al. (2015), used to describe the interaction between spatial rainfall resolution and model scale

| $\Delta s$ | Catchment sampling number | | | Runoff sampling number | | | Sewer sampling number | | |
|---|---|---|---|---|---|---|---|---|---|
| | SD1 | SD2 | FD | SD1 | SD2 | FD | SD1 | SD2 | FD |
| 100 m | 0.03 | 0.04 | 0.04 | 0.25 | 2.29 | 10 | 0.19 | 1.73 | 1.73 |
| 500 m | 0.17 | 0.20 | 0.20 | 1.23 | 11.47 | 50 | 0.94 | 8.65 | 8.65 |
| 1000 m | 0.34 | 0.40 | 0.40 | 2.45 | 22.94 | 100 | 1.87 | 17.30 | 17.30 |
| 3000 m | 1.03 | 1.20 | 1.20 | 7.35 | 68.82 | 300 | 5.62 | 51.91 | 51.91 |