# Peer review of "Critical scales to explain urban hydrological response"

_Hydrology and Earth System Sciences, 2017_

## Short Comment (SC1) · 8 Feb 2018

How do you justify the use of rainfall data from the Netherlands (p. 3, line 23) applied to models of a catchment in the UK (p. 3, line 3)? Since the aim of the study is to understand the effect of rainfall aggregation, this may be OK, but, in my opinion, the discrepancy in location should at least be noted and potential effects on results discussed. Urban designs, soils, and landscapes are influenced by rainfall patterns, so the drainage network of the Cranbrook catchment is designed for rainfall patterns expected in London and not the Netherlands. This could potentially have some influence on the specific results of the study. It may be that rainfall in London and the Netherlands is sufficient similar that it doesn't matter, but for readers not very familiar with the meteorology of Europe this isn't apparent.

---

## Referee Comment (RC1) · K. Förster (Referee) · 9 Feb 2018

**General comments:**

In this research article, the authors present a comprehensive study on the effects of different levels of temporal and spatial rainfall aggregation on the hydrological response derived by three different hydrological models. These models were set-up for an urban hydrological system in Cranbrook/London and cover a broad range in terms of spatial discretization and complexity. X-band radar data from the Netherlands with high resolution in space and time was used to perform the model experiments. In a very comprehensive study, the authors apply a lot of measures to shed led on rainfall and model scale. While most of these measures are taken from literature, the authors introduce three new scaling factors which might be helpful in selecting appropriate scales. In my opinion, this paper is written very well, the experiments are sound and the results are valid. The topic fits very well into the journal's scope and the results are of great interest to the community. However, in my opinion, the paper needs a few minor revisions in order to improve the readability. Some ideas might help you to strengthen the thoughts and ideas of the paper:

- As pointed out earlier by Anne Jefferson, it is very important to mention that the X-band radar in Cabauw is used to perform sensitivity studies rather than reconstructing past events. X-band radars are capable of observing precipitation with a high spatial and temporal resolution. In terms of scale, their spacing and support is high. However, their extent is small. You should at least mention in Section 2.2 that the rainfall data does not represent events observed in Cranbrook. Later, in the Conclusions Section you should critically review this issue.

- You introduce a lot of scaling measures. For the reader, it is sometimes difficult to keep everything in mind. While reading the manuscript, especially Section 4, I needed to review the methodological framework several times (sometimes I really felt lost by the symbols which are not so common). Moreover, you introduce dimensionless values based on rainfall and model characteristics. In order to better understand, how units cancel out, a small table or a list of symbols (including units) would be really helpful to improve the understandability of the manuscript if possible.

- When reading the title, I expected to read something about a framework that deals with finding best scales for rainfall input data and model resolution / complexity. The paper focus on this in a very comprehensive way based on a good structure and nice explanations. However, I feel that the paper ends up at a point when it would be most interesting for the community: How do these alpha values help us to select a specific rainfall resolution in terms of critical scales (as I would expect

it from the title)? From the manuscript I couldn't get any information regarding this question. In my opinion, even though the title reflects the content, it seems to me a little bit too general given that this specific case study does not allow to draw any conclusion about appropriate values which might be transferred to other settings. However, the methodology is for sure of great value and should be applied to other sites in the future.

- In this context, another idea might be interesting: When working with observed rainfall (even with station data; unfortunately, in most cases X-band radar is not available), we are faced with the situation that we only have a few stations with minute-scale rainfall data (especially if we would like to analyze events that happened several years ago). In reality, we cannot work with X-band radars located hundreds of kilometers away from out site of interest. Therefore, if we consider some station data, it might be feasible to apply the highest temporal resolution available. Then the question remains: What resolution / complexity of the model would be best suitable given that rainfall is restricted to a fixed interval? You figured out and mentioned that the impact of the model is smaller than the impact of rainfall. However, I was wondering if it would be worth at least to briefly consider the opposite question as well?

**Specific comments:**

P2L13-14: I am not sure if this is true. In principle, the sensitivity of hydrological models Is understood very well, even though it is not always reported in a quantitative way. I would suggest rephrasing this sentence. You could argue that the interactions of scales (rainfall and models) requires some more attention. Please also consider adding a reference as well.

P4L7: What means "climatological" in this context? Is this variogram constructed using a data of multi-year period? I think that nine events are a too small number to state that it is a climatology.

P5L2: In my opinion, rainfall velocity is ambiguous (velocity of raindrops vs. storm motion). Do you mean storm motion velocity? Please consider rephrasing.

P5L25: Is rainfall cluster dimension the SZ value which is introduced later? Please consider defining it here.

P8L10: Do you mean the R values? I would expect P values provided as rainfall intensity.

P8L18: Does parallel mean in upstream or downstream direction? As far as I understand your explanations, this information is missing.

P15L31: Here it would be helpful to add the meaning of each symbol in parentheses.

P16L4-8: Here, you should also refer to Figure 12(a).

P17L20: Here, it would be interesting to address the opposite question as well (what model setting would be best suitable for a given rainfall spatial and temporal scale).

**Technical corrections:**

P2L1: led?

P2L16 the sensitivity

P3L19: models (plural)

P4L24: in a basin?

P5L23: pixels (plural form)

P6L27: Please consider replacing bigger by larger. The same might apply to the next line.

P8L29: Reference to Section 3.3?

P9L8: Kolmogorov's theory

P10L28: as well as the required

P16L13: plot

P25: In the caption of Figure 3 (first line) it would be helpful to mention that the temporal percentage is shown in b,e,h,k in order to underline the difference between the first and the second column.

---

## Referee Comment (RC2) · Anonymous Referee #2 · 13 Feb 2018

General comments:

The authors analyse rainfall variability in space and time in relation to catchment characteristics and model complexity. They use various indices to characterise these variability. Beside known indices they introduced some new indices and a new classification of rainfall variability based on percentage of coverage above a selected threshold. For the analyses data from nine rainfall events observed with a X–Band radar located in the Netherlands are utilised. For modelling three hydrologic/ hydraulic models with different complexity and a sewage network in London are applied. The results show that the new classification allows a good representation of the storm cores and gives information about the required scales for hydrological modelling.

The paper is quite well written and clear in structure. An interesting innovation is seen

in the new classification of rainfall events. The conclusions are supported by the analyses. However, one problem is the readability of the article due to the large number of specific indices which are usually not very common. Many times a had to leaf back to the methodology section and re-read the definitions of the indices to understand the discussion and conclusions. I don't have a real good idea how to improve the readability regarding this issue; one possibility would be to append an extended table of symbols with short definitions including ranges of the indices; another possibility would be to reduce the number of indices. Otherwise there are only a few minor comments for improvement (see below). Altogether the paper is very interesting and well worth of publication after the authors have the opportunity do some revisions.

Detailed comments:

1. Page 3: The location for rainfall data observation (Netherlands) and analysed sewage networks (London) don't correspond. Please, explicitly state this mismatch and include a brief discussion why you have chosen this setting.

2. Page 4, line 12: What are the left and right boundaries of the area under the variogram?

3. Page 4, line 13: Why "correlogram"? You probably mean here also the variogram.

4. Page 10, lines 28ff: Is it really the case that the spatial variability index is increasing for storms with a large range? It looks like the opposite in Table 3 (e.g. E2, E4, E6,E8).

5. Page 36, Table 2: E2 has the same starting and ending times as E1?

6. Page 36, Table 2: I do not understand what min and max means in the column with total depth (over time total depth cannot have min and max; is this regarding different spatial extents)?

---

## Author Comment (AC1) · 26 Feb 2018

Dear Anne Jefferson,

Thank you for pointing out the issue of representative climatology. As you mentioned, the 2 locations present quite similar climatological characteristics and, for this reason, it was reasonable possible to assume that rainfall events measured in The Netherlands are realistic also for the London area. However, as suggested, I'll add a paragraph in the Data section explaining better how we made this assumption and which effects this can have on the results.

---

## Author Comment (AC2) · 26 Feb 2018

Dear Kristian Förster,

Thank you for comments and suggestions. Below you can find the reply [AR] to your comments [RC].

General comments:

[RC] As pointed out earlier by Anne Jefferson, it is very important to mention that the X-band radar in Cabauw is used to perform sensitivity studies rather than reconstructing past events. X-band radars are capable of observing precipitation with a high spatial and temporal resolution. In terms of scale, their spacing and support is high. However, their extent is small. You should at least mention in Section 2.2 that the rainfall data

does not represent events observed in Cranbrook. Later, in the Conclusions Section you should critically review this issue.

[AR] As replied also to Anne Jefferson, we will add a paragraph to explain better the assumptions we made to apply data measured in the Netherlands to the hydrological model in the London area and to describe the possible effects that can derive from this choice.

[RC] You introduce a lot of scaling measures. For the reader, it is sometimes difficult to keep everything in mind. While reading the manuscript, especially Section 4, I needed to review the methodological framework several times (sometimes I really felt lost by the symbols which are not so common). Moreover, you introduce dimensionless values based on rainfall and model characteristics. In order to better understand, how units cancel out, a small table or a list of symbols (including units) would be really helpful to improve the understandability of the manuscript if possible.

[AR] Thank you for the suggestion. We will add a table with list of symbols used, units and a small description of the parameters to help the reader follow the manuscript.

[RC] When reading the title, I expected to read something about a framework that deals with finding best scales for rainfall input data and model resolution / complexity. The paper focus on this in a very comprehensive way based on a good structure and nice explanations. However, I feel that the paper ends up at a point when it would be most interesting for the community: How do these alpha values help us to select a specific rainfall resolution in terms of critical scales (as I would expect it from the title)? From the manuscript I couldn't get any information regarding this question. In my opinion, even though the title reflects the content, it seems to me a little bit too general given that this specific case study does not allow to draw any conclusion about appropriate values which might be transferred to other settings. However, the methodology is for sure of great value and should be applied to other sites in the future.

[AR] We will consider to change the title or part of the conclusions in order to highlight

the fact that this work refers to a specific study case, with particular characteristics. However, specific critical scale are derived, in association with the alpha values. A paragraph will be added in order to explain better how the alpha values help us to select the rainfall resolution required.

[RC] In this context, another idea might be interesting: When working with observed rainfall (even with station data; unfortunately, in most cases X-band radar is not available), we are faced with the situation that we only have a few stations with minute-scale rainfall data (especially if we would like to analyze events that happened several years ago). In reality, we cannot work with X-band radars located hundreds of kilometers away from out site of interest. Therefore, if we consider some station data, it might be feasible to apply the highest temporal resolution available. Then the question remains: What resolution / complexity of the model would be best suitable given that rainfall is restricted to a fixed interval? You figured out and mentioned that the impact of the model is smaller than the impact of rainfall. However, I was wondering if it would be worth at least to briefly consider the opposite question as well?

[AR] Thank you for the suggestion. It will indeed be interesting to investigate what model complexity is reasonable when high-resolution rainfall data are not available. We will add a brief comment about what we learn in this respect from our results, based on 3 different model complexities and add suggestions in the future steps section.

Specific comments:

[RC] P2L13-14: I am not sure if this is true. In principle, the sensitivity of hydrological models Is understood very well, even though it is not always reported in a quantitative way. I would suggest rephrasing this sentence. You could argue that the interactions of scales (rainfall and models) requires some more attention. Please also consider adding a reference as well. P4L7: What means "climatological" in this context? Is this variogram constructed using a data of multi-year period? I think that nine events are a too small number to state that it is a climatology. P5L2: In my opinion, rainfall velocity

is ambiguous (velocity of raindrops vs. storm motion). Do you mean storm motion velocity? Please consider rephrasing. P5L25: Is rainfall cluster dimension the SZ value which is introduced later? Please consider defining it here. P8L10: Do you mean the R values? I would expect P values provided as rainfall intensity. P8L18: Does parallel mean in upstream or downstream direction? As far as I understand your explanations, this information is missing. P15L31: Here it would be helpful to add the meaning of each symbol in parentheses. P16L4-8: Here, you should also refer to Figure 12(a). P17L20: Here, it would be interesting to address the opposite question as well (what model setting would be best suitable for a given rainfall spatial and temporal scale).

[AR] Thank you for these comments, we will consider rephrasing unclear sentences, adding a more complete explanation where requested in order to increase the readability of the manuscript and avoid misunderstanding. Proper references will be added where needed.

Technical corrections [AR] Thank you for highlight typos and writing errors. They will be corrected in the new version of the manuscript.

---

## Author Comment (AC3) · 2 Mar 2018

Dear reviewer,

Thank you for comments and suggestions. Below you can find the reply [AR] to your comments [RC].

[RC]General comments: The authors analyse rainfall variability in space and time in relation to catchment characteristics and model complexity. They use various indices to characterise these variability. Beside known indices they introduced some new indices and a new classification of rainfall variability based on percentage of coverage above a selected threshold. For the analyses data from nine rainfall events observed with a X–Band radar located in the Netherlands are utilised. For modelling three hydrologic/

hydraulic models with different complexity and a sewage network in London are applied. The results show that the new classification allows a good representation of the storm cores and gives information about the required scales for hydrological modelling. The paper is quite well written and clear in structure. An interesting innovation is seen in the new classification of rainfall events. The conclusions are supported by the analyses. However, one problem is the readability of the article due to the large number of specific indices which are usually not very common. Many times a had to leaf back to the methodology section and re-read the definitions of the indices to understand the discussion and conclusions. I don't have a real good idea how to improve the readability regarding this issue; one possibility would be to append an extended table of symbols with short definitions including ranges of the indices; another possibility would be to reduce the number of indices. Otherwise there are only a few minor comments for improvement (see below). Altogether the paper is very interesting and well worth of publication after the authors have the opportunity do some revisions.

[AR] Regarding the readability of the paper, we will add a table to help the reader with the new parameters, adding symbols, names, descriptions and units.

[RC] Detailed comments: 1. Page 3: The location for rainfall data observation (Netherlands) and analysed sewage networks (London) don't correspond. Please, explicitly state this mismatch and include a brief discussion why you have chosen this setting.

[AR] Thank you for pointing this out. We agree that the reason why we chose to apply data measured in the Netherlands over the London area was not well explained in the manuscript. We will add a paragraph motivating the choice of using rainfall datasets from one and hydrological model from another, albeit same climatological, region in the Data section and some comments about the possible consequences of this assumption

[RC]2. Page 4, line 12: What are the left and right boundaries of the area under the variogram? 3. Page 4, line 13: Why "correlogram"? You probably mean here also the variogram. 4. Page 10, lines 28: Is it really the case that the spatial variability index is
increasing for storms with a large range? It looks like the opposite in Table 3 (e.g. E2, E4, E6,E8).

[AR] We will clarify the questions about the variogram clarified and rephrase sentences they may have caused confusion.

[RC] 5. Page 36, Table 2: E2 has the same starting and ending times as E1?

[AR] Yes, it has. E1 and E2 are part of the same event, yet they represent different rainfall cells, hence different pixel regions of the radar measurement.

[RC]6. Page 36, Table 2: I do not understand what min and max means in the column with total depth (over time total depth cannot have min and max; is this regarding different spatial extents)?

[AR] The total depth is for each pixel, so mean and max refer to different pixel. A better description of this parameter will be added in the manuscript and in the table.

---

## Author Response (AR1)

**Reply to Reviewer #1, Kristian Foster**

**General comments:**

In this research article, the authors present a comprehensive study on the effects of different levels of temporal and spatial rainfall aggregation on the hydrological response derived by three different hydrological models. These models were set-up for an urban hydrological system in Cranbrook/London and cover a broad range in terms of spatial discretization and complexity. X-band radar data from the Netherlands with high resolution in space and time was used to perform the model experiments. In a very comprehensive study, the authors apply a lot of measures to shed led on rainfall and model scale. While most of these measures are taken from literature, the authors introduce three new scaling factors which might be helpful in selecting appropriate scales. In my opinion, this paper is written very well, the experiments are sound and the results are valid. The topic fits very well into the journal's scope and the results are of great interest to the community. However, in my opinion, the paper needs a few minor revisions in order to improve the readability. Some ideas might help you to strengthen the thoughts and ideas of the paper:

- As pointed out earlier by Anne Jefferson, it is very important to mention that the X-band radar in Cabauw is used to perform sensitivity studies rather than reconstructing past events. X-band radars are capable of observing precipitation with a high spatial and temporal resolution. In terms of scale, their spacing and support is high. However, their extent is small. You should at least mention in Section 2.2 that the rainfall data does not represent events observed in Cranbrook. Later, in the Conclusions Section you should critically review this issue.

[AR1] The following text was added to better explain the assumption of applying rainfall data from the Netherlands to Cranbrook, located in the London area:

- Section 2.2, Rainfall data, starting of the paragraph: "Cranbrook was chosen for this study because of the availability of high quality models at different spatial resolutions. However, for this study area, only low-resolution rainfall data were available. For this reason, rainfall events measured at a different location, with similar climatological characteristics, were synthetically applied over the Cranbrook catchment. Rainfall events were selected from a dataset collected by a dual polarimetric X-Band weather radar located in Cabauw (CAESAR weather station, NL), considering that the Netherland and United Kingdom are both in the European temperate oceanic climate (Cfb, following the Koppen classification)."

- Conclusions, first paragraph: "Rainfall data measured at 100 m and 1 min resolution by a dual polarimetric X-band radar located in the Netherlands, , were aggregated to obtain different rainfall resolutions and then used as input for the hydrological models. Storm events were assumed to be representative of the rainfall regime in the London area, as London and Cabauw are situated in the same (Temperate oceanic) climatological region."

- Conclusion, last paragraph, future steps: "Rainfall events measured directly over the study area should be evaluated to allow a proper comparison between model results and observations. In particular, using local rainfall data as input for the model combined with local discharge measurements, would enable direct investigation the sensitivity of the hydrological response with respective to an observed reference."

- You introduce a lot of scaling measures. For the reader, it is sometimes difficult to keep everything in mind. While reading the manuscript, especially Section 4, I needed to review the methodological framework several times (sometimes I really felt lost by the symbols which are not so common). Moreover, you introduce dimensionless values based on rainfall and model characteristics. In order to better understand, how units cancel out, a small table or a list of symbols (including units) would be really helpful to improve the understandability of the manuscript if possible.

[AR2] As suggested, a table that summarises all symbols, was added to the manuscript. Hopefully this will help the reader understanding the manuscript. The table was added at the beginning of Section 3. Methods, introduced by the following text: "Table 3 presents the list of symbols and abbreviations used in this work."

- When reading the title, I expected to read something about a framework that deals with finding best scales for rainfall input data and model resolution / complexity. The paper focus on this in a very comprehensive way based on a good structure and nice explanations. However, I feel that the paper ends up at a point when it would be most interesting for the community: How do these alpha values help us to select a specific rainfall resolution in terms of critical scales

(as I would expect it from the title)? From the manuscript I couldn't get any information regarding this question. In my opinion, even though the title reflects the content, it seems to me a little bit too general given that this specific case study does not allow to draw any conclusion about appropriate values which might be transferred to other settings. However, the methodology is for sure of great value and should be applied to other sites in the future.

[AR3] Although we were indeed quite encouraged by the results, we do not feel comfortable generalizing the results beyond the present case. To highlight the specificity of this work the title is changed to : "Critical scales to explain urban hydrological response: an application in Cranbrook, London" and the following sentence is added to the conclusions: "Results presented in this paper are related to one specific study case and need further investigations, based on cases in different climatological regions and with different hydrological characteristics to test to what extent they can be generalised."

- In this context, another idea might be interesting: When working with observed rainfall (even with station data; unfortunately, in most cases X-band radar is not available), we are faced with the situation that we only have a few stations with minute-scale rainfall data (especially if we would like to analyze events that happened several years ago). In reality, we cannot work with X-band radars located hundreds of kilometers away from out site of interest. Therefore, if we consider some station data, it might be feasible to apply the highest temporal resolution available. Then the question remains: What resolution / complexity of the model would be best suitable given that rainfall is restricted to a fixed interval? You figured out and mentioned that the impact of the model is smaller than the impact of rainfall. However, I was wondering if it would be worth at least to briefly consider the opposite question as well?

[AR4] Thanks for this suggestion. The question of suitable model complexity and resolution given a certain rainfall input is not easy to answer, as it is strongly dependent on the application context. Our study showed that in terms of sensitivity, applying complex, high resolution models in combination with low resolution rainfall input results in similar performance (or lack thereof) as applying a low-complexity model. However, further investigations are needed to evaluate model suitability in the context of observed data.

**Specific comments:**

[RC] P2L13-14: I am not sure if this is true. In principle, the sensitivity of hydrological models Is understood very well, even though it is not always reported in a quantitative way. I would suggest rephrasing this sentence. You could argue that the interactions of scales (rainfall and models) requires some more attention. Please also consider adding a reference as well.

[AR5] The sentence was rephrased as follows: "However, sensitivity of hydrological models at different rainfall and catchments scale and the interaction between rainfall and catchment variability need a deeper investigation (Ochoa-Rodriguez et al., 2015; Pina et al., 2016; Cristiano et al., 2017)."

[RC] P4L7: What means "climatological" in this context? Is this variogram constructed using a data of multi-year period? I think that nine events are a too small number to state that it is a climatology.

[AR6] The term "climatological" is generally used in the field to refer to rainfall based variograms (Bastin, 1984).

[RC] P5L2: In my opinion, rainfall velocity is ambiguous (velocity of raindrops vs. storm motion). Do you mean storm motion velocity? Please consider rephrasing.

[AR7] In the paper we indeed use the term "velocity" in the sense of storm motion velocity. To avoid misunderstanding, the term velocity was replaced with "storm motion velocity".

[RC] P5L25: Is rainfall cluster dimension the SZ value which is introduced later? Please consider defining it here.

[AR8] Indeed, it refers to $S_Z$. The text was rephrased as follows, to introduce $S_Z$:" To analyse the spatial variability of the storm core, we identified, for each rainfall event, the main rainfall cluster dimension $S_Z$ above the selected thresholds Z, as defined in Section 3.1.4."

[RC] P8L10: Do you mean the R values? I would expect P values provided as rainfall intensity.

[AR9] P indicates the rainfall peak, while R indicates the rainfall intensity. To define the coefficient of determination, R values are required and not the peak. R is defined previously in the manuscript as rainfall intensity and it is also included in the new table with the list of symbols.

[RC] P8L18: Does parallel mean in upstream or downstream direction? As far as I understand your explanations, this information is missing.

[AR10] Parallel in downstream direction. This information was added to the text and the sentence was rephrased as follows: "… the storm movement main direction was parallel to the main downstream direction of flow in pipes…"

[RC] P15L31: Here it would be helpful to add the meaning of each symbol in parentheses.

[AR11] The paragraph was rephrased as follows: "Higher values of the scaling factors $\theta_S$ (ratio between minimum required spatial resolution and rainfall spatial resolution), $\theta_T$ (ratio between minimum required temporal resolution and rainfall temporal resolution) and $\theta_{ST}$ (combination of spatial and temporal scaling factors) are generally associated with higher modelling performance, expressed in terms of $R^2$."

[RC] P16L4-8: Here, you should also refer to Figure 12(a).

[AR12] Reference to Fig.12(a) was added.

[RC] P17L20: Here, it would be interesting to address the opposite question as well (what model setting would be best suitable for a given rainfall spatial and temporal scale).

[AR13] See also response to similar remark made earlier, [AR4]

**Technical corrections:**

[RC] P2L1: led?

[AR14] Corrected

[RC]P2L16 the sensitivity

[AR15] Corrected

[RC]P3L19: models (plural)

[AR16] Corrected

[RC]P4L24: in a basin?

[AR17] Corrected with "… in the basin."

[RC]P5L23: pixels (plural form)

[AR18] Corrected

[RC]P6L27: Please consider replacing bigger by larger. The same might apply to the next line.

[AR19] Corrected

[RC]P8L29: Reference to Section 3.3?

[AR20] Corrected

[RC]P9L8: Kolmogorov's theory

[AR21] Corrected

[RC]P10L28: as well as the required

[AR22] Corrected

[RC]P16L13: plot

[AR23] Corrected

[RC]P25: In the caption of Figure 3 (first line) it would be helpful to mention that the temporal percentage is shown in b,e,h,k in order to underline the difference between the first and the second column.

[AR24] The following text was added: "Temporal percentage of coverage above the selected threshold, defined as number of time steps above the threshold at each pixel, divided by the total duration of the event (b, e, h, k). Temporal percentage is presented for each rainfall event and the number above each boxplot indicates the total duration of the rainfall event."

**Reply to Reviewer #2**

**General comments:**

[RC] The authors analyse rainfall variability in space and time in relation to catchment characteristics and model complexity. They use various indices to characterise these variability. Beside known indices they introduced some new indices and a new classification of rainfall variability based on percentage of coverage above a selected threshold. For the analyses data from nine rainfall events observed with a X–Band radar located in the Netherlands are utilised. For modelling three hydrologic/ hydraulic models with different complexity and a sewage network in London are applied. The results show that the new classification allows a good representation of the storm cores and gives information about the required scales for hydrological modelling.

The paper is quite well written and clear in structure. An interesting innovation is seen in the new classification of rainfall events. The conclusions are supported by the analyses. However, one problem is the readability of the article due to the large number of specific indices which are usually not very common. Many times a had to leaf back to the methodology section and re-read the definitions of the indices to understand the discussion and conclusions. I don't have a real good idea how to improve the readability regarding this issue; one possibility would be to append an extended table of symbols with short definitions including ranges of the indices; another possibility would be to reduce the number of indices. Otherwise there are only a few minor comments for improvement (see below). Altogether the paper is very interesting and well worth of publication after the authors have the opportunity do some revisions.

[AR] As suggested, a table that summarises all symbols, was added to the manuscript. Hopefully this will help the reader understanding the manuscript. The table was added at the beginning of Section 3. Methods, introduced by the following text: "Table 3 presents the list of symbols and abbreviations used in this work."

**Detailed comments:**

[RC] 1. Page 3: The location for rainfall data observation (Netherlands) and analysed sewage networks (London) don't correspond. Please, explicitly state this mismatch and include a brief discussion why you have chosen this setting.

[AR] The following text was added to better explain the assumption of applying rainfall data from the Netherland in London:

[RC] 2. Page 4, line 12: What are the left and right boundaries of the area under the variogram? 3. Page 4, line 13: Why "correlogram"? You probably mean here also the variogram.

[AR] As explained in Ochoa-Rodriguez et al. (2015) , the integral over the entire domain R. The correct term to use in this case is correlogram. Considering the misunderstandings and ambiguities due to the short summary of part Ochoa-Rodriguez et al. (2015) work in this manuscript, we highlighted the reference to Ochoa-Rodriguez et al. (2015), adding the following text at the beginning of the paragraph: "We computed characteristics spatial scale based on a climatological variogram, following the approach outlined by Ochoa-Rodriguez et al. (2015)."

[RC] 4. Page 10, lines 28ff: Is it really the case that the spatial variability index is increasing for storms with a large range? It looks like the opposite in Table 3 (e.g. E2, E4, E6,E8).

[AR] If we consider only storm with large (>2500 m) spatial range, when the spatial range increases (E4 = 3219 m, E8 = 3644 m, E3 = 4655 m) , we observe a general increase of the spatial rainfall variability index (E4 = 1.5 mm\h, E8 = 4.2 mm\h, E3 = 6.5 mm\h) . This is, as we said a general trend, with some small exceptions.

[RC] 5. Page 36, Table 2: E2 has the same starting and ending times as E1?

[AR] Yes, it has. E1 and E2 are part of the same event, yet they represent different rainfall cells, hence different pixel regions of the radar measurement.

[RC] 6. Page 36, Table 2: I do not understand what min and max means in the column with total depth (over time total depth cannot have min and max; is this regarding different spatial extents)?
[AR] The minimum and maximum values of the total depth were referred to the individual pixels, while the average is over the

total area. To avoid this misunderstanding, the name of the column has been changed into: "Total depth, (areal average / pixel min / pixel max)".

The increase of high-resolution topographical data availability  led to a development of different types of hydrological models (Mayer, 1999; Fonstad et al., 2013; Tokarczyk et al., 2015). These models represent spatial variability of catchments in several ways, varying from lumped systems, where spatial variability is averaged into sub-catchments, to distributed models, which evaluate the variability dividing the basin with a mesh of interconnected elements based on elevation (Zoppou, 2000; Fletcher et al., 2013; Pina et al., 2014; Salvadore et al., 2015). Salvadore et al. (2015) analysed the most used hydrological models, comparing different model complexities and approaches. An investigation of the differences between high resolution semi and fully distributed models was proposed by Pina et al. (2016), where flow patterns generated with different model types were studied and compared to observations. This work suggested that although fully distributed models allow to represent catchment variability in space in a more realistic way, they did not lead to the best modelling results because the operation of this type of models requires very high quality and resolution data, including rainfall input.

Both rainfall and model resolution and scale are expected to have strong effects on hydrological response sensitivity. An increase of sensitivity is expected for small drainage areas and for rainfall events with high variability in space and time. Sensitivity to rainfall data resolution generally increases for smaller urban catchments. However, sensitivity of hydrological models at different rainfall and catchments scale  and the interaction between rainfall and catchment variability need a deeper investigation (Ochoa-Rodriguez et al., 2015; Pina et al., 2016; Cristiano et al., 2017). This work builds upon Ochoa-Rodriguez et al. (2015), who showed that the influence of rainfall input resolution decreases with the increase catchment area and that the interaction between spatial and temporal rainfall resolution is quite strong. We investigate the sensitivity of urban hydrological response to different rainfall and catchment scales, with the aim of answering the following research questions:

- How should rainfall variability in space and time be classified?

- How does small scale rainfall variability affect hydrological response in a highly urbanized area?

- How does model complexity affect sensitivity of model outcomes to rainfall variability?

- How does the relationship between storm scale and basin scale affect hydrological response?

The paper is structured as follows. Section 2 presents the case study, describing study area, models and rainfall data used in this work. Methodology applied to identify variability in space and time of model and rainfall and hydrological analysis are explained in Section 3. Section 4 presents the results connected to the model and rainfall variability analysis and to the hydrological analysis respectively. In Section 5, results are discussed, by comparing the influence of rainfall and model characteristics and identifying dimensionless parameters to describe the relation between rainfall and model scale and rainfall resolution used. Conclusions and future steps are presented in the last section.

**2 Pilot catchment and datasets**

**2.1 Study area and available models**

The city of London (UK) is exposed to high pluvial flood risk in the last years. The Cranbrook catchment, in the London Borough of Redbridge, is a densely urbanized residential area. For this reason, it has been chosen as study area. A total area of approximately 860 ha is connected to the drainage network, and rainfall is drained with a separate sewer system.

For this small catchment, several urban hydrodynamical models have been set up in InfoWorks ICM (Innovyze, 2014). Three models with different representations of surface spatial variability, are used in this study: SD1 - Simplified semi-distributed low resolution, SD2 - Semi-distributed high resolution and FD - Fully distributed 2D high resolution.

Table 3 summarises the main characteristics of the three models: number of nodes, pipes and sub-catchments, dimensions of subacatchments, two dimensional surface elements and degree of imperviousness. The first model, SD1, is a low resolution semi-distributed model, initially setup by the water utility (Thames Water) back in 2010 to gain a strategic understanding of the catchment. This model divides the area into 51 sub-catchments, connected with 242 nodes and 270 pipes, for a total drainage network length of just over 15 km. The other two models, SD2 and FD, have been developed at Imperial College London (Simões et al., 2015; Wang et al., 2015; Ochoa-Rodriguez et al., 2015; Pina et al., 2016). SD2 and FD share the same sewer network design(6963 nodes and 6993 pipes), but use different surface representations. In SD2 the drainage area is divided into 4409 sub-catchments, where rainfall runoff processes are modelled in a lumped way and wherein rainfall is assumed to be uniform. In FD, instead, the surface is modelled with a dense triangular mesh (over 100'000 elements), based on a high resolution (1 m x 1 m) Digital Terrain Model (DTM). The rainfall - runoff transformation is different for the two types of models. For SD2, runoff volumes are estimated from rainfall depending on the land use type and routed, while for FD, runoff volumes are estimated and applied directly on the two-dimensional elements of the overland surface. Figure 1 illustrates how the surface area is modelled for each of the three models and sewer networks.

**2.2 Rainfall data**

Cranbrook was chosen for this study for the interesting availability of models at different spatial resolutions. However, for this study area, only low-resolution rainfall data were available. For this reason, rainfall events were measured in a different location, with similar climatological characteristics, and then synthetically applied over the catchment. Rainfall events were selected from a dataset collected by a dual polarimetric X-Band weather radar located in Cabauw (CAESAR weather station, NL), assuming that Dutch and British storm events present similar climatological characteristics. For technical specifications of the X-band radar device see Ochoa-Rodriguez et al. (2015). The selected events were measured with a resolution of 100 m x 100 m in space and 1 min in time, much higher than what is obtained with conventional radar networks (1000 m x 1000 m and 5 min). Rainfall data where applied to the Cranbrook catchment, using sixteen combinations of space and time resolution aggregated from the 100 m - 1 min resolution: four spatial resolutions, $\Delta s$, (100 m, 500 m, 1000 m and 3000 m) with four temporal resolutions, $\Delta t$, (1 min, 3 min, 5 min and 10 min) (see Ochoa-Rodriguez et al. (2015) for a motivation of the different

resolution combinations). Nine rainfall events, measured between January 2011 and May 2014, were used as model input in this study. Storm characteristics are presented in Table 4.

**3 Methods**

**3.1**

5 In this section, different ways of classifying spatial and temporal rainfall scale are described, as well as some possible classification of catchment characteristics. We propose a new characterization of spatial and temporal rainfall variability, based on percentage of coverage above selected thresholds. Table 1 presents the list of symbols and abbreviations used in this work.

**3.1 Characterizing storms' spatial and temporal rainfall scale**

[revised manuscript text omitted]
$ (ratio between minimum required spatial resolution and rainfall spatial resolution), $\theta_T$ and (ratio between minimum required temporal resolution and rainfall temporal resolution) and $\theta_{ST}$ (combination of spatial and temporal scaling factors) are generally associated with higher modelling performance, expressed in terms of $R^2$. The combined spatial-temporal scaling factor, $\theta_{ST}$, in particular indicates

10   how high $R_Q^2$ values are obtained for $\theta_{ST} > 15$ ($R^2 > 0.9$).

As discussed in Section 4.4.3, both rainfall scale and catchment characteristics strongly affect sensitivity of hydrological response to rainfall resolution. For this reason, the new dimensionless factors proposed combine rainfall and catchment properties. From results shown in Fig. 11(a-c), spatial variability seems to have a better relation with the sensitivity variability than the temporal scale . The and, for this reason, the factor $\alpha_1$ takes into account only especially focuses on the spatial scale of model

15   and rainfall variability. Figure 11(d) and Fig. 12(a) shows $R_Q^2$ as a function of $\alpha1$. The plot presents a clear trend, indicating low model performance for low values of $\alpha_1$ and high performance for values of $\alpha_1$ larger than 100.

Figure 11(e) shows $\alpha_2$ and response sensitivity. For values of $\alpha_2 > 40$, $R_Q^2$ is higher than 0.95, indicating a very good performance. For values of $\alpha_2 < 10$, $R_Q^2$ is lower than 0.8. Figure 12(b) shows the same plot on a logarithmic scale, which better visualises thresholds of performance. Different resolutions are highlighted in the plot. Low resolution in space generally

20   lead to a lower $\alpha$'s values than low temporal resolution, and consequently to a lower performance of the model.

Figure 11(f) and Fig. 12(c) plots plot $R_Q^2$ against $\alpha_3$. Figure 12(c) indicates that for values of $\alpha_3$ higher than 3000 a high performance of $R_Q^2$ is guaranteed ($R^2 > 0.90$). For $400 < \alpha_3 < 3000$ the performance of $R_Q^2$ drops to 0.8.

Comparing the scaling factors, we observe that $\alpha_2$ works better in distinguishing critical resolutions for a given model performance. There are indeed, less points with high $R_Q^2$ below the identified thresholds. Moreover, $\alpha_2$ should be preferred

25   because it allows to use fewer parameters, without losing information about temporal characteristics, as it is for $\alpha_1$.

**5   Conclusions**

In this study we investigated effects of rainfall and catchment scales on sensitivity of urban hydrological models to different rainfall input resolutions. The aim was to identify dimensionless ratios of storm and catchment scales that support critical resolution for reproducing hydrological response. Cranbrook, a small urbanized area of 861 ha, was analysed with the help

30   of two semi-distributed and a fully distributed models. Rainfall data measured by a dual polarimetric X-band radar located in the Netherlands, at 100 m and 1 min resolution, were aggregated to obtain different rainfall resolutions and then used as input for the hydrological models, assuming that Dutch and British rainfall events have similar climatological characteristics.

A new rainfall classification method, based on cluster identification was presented in this work. Different rainfall classification methods were used to characterize storm event scales.

From this work we draw the following conclusions.

- Rainfall classification based on clustering is an easy and fast method to quantify spatial scale of rainfall events. In particular, rainfall clusters associated with the 75%-ile threshold turned out to give a realistic approximation of the spatial dimension of the storm core.

- Spatial and temporal aggregation of rainfall data can have a strong effect on rainfall peak and intensity. Rainfall peaks were reduced up to 80% when aggregating in space to 3000 m resolution or in time at 10 min resolution. Both space and time have a strong influence on peak attenuation. Temporal aggregation has a stronger influence at 1 - 5 min resolution, while aggregation in space has bigger impact at low (1000 - 3000 m) resolution.

- Lag time estimated for the investigated sub-catchments was used to represent the temporal characteristics of models. Lag time increased with the catchment area size, yet varied strongly between events (approx. by a factor of 2, 25-75%-ile range). Mean lag time fitted an empirical power law similar to the one proposed by Berne et al. (2004), yet with a higher intercept.

- Effects of rainfall aggregation in space and time on hydrological response depend on rainfall event characteristics. Rainfall events with constant intensity are less affected by aggregation than small scale intermittent events. However, results showed that aggregation effects are stronger for rainfall than flow. Results showed that smoothing of rainfall peak intensities by aggregation was much stronger than for flows. Rainfall aggregation effects on hydrological response are smoothed during the rainfall runoff transformation processes.

- For the case study under consideration, model spatial resolution does not appear to have a big impact on hydrological response sensitivity to rainfall input resolution. Three models of different complexity were all sensitive to rainfall resolution. Low resolution model was more sensitive to rainfall resolution for small scale storms, while the high resolution fully distributed model showed stronger sensitivity at larger catchment scale.

- Rainfall and catchment scales were shown to have a strong impact on hydrological response sensitivity. This indicates that the relation between rainfall and catchment scale needs to be taken into account when investigating the hydrological response of a system.

- New spatial, temporal and combined scaling factors were introduced to analyse hydrological response sensitivity to rainfall resolution. These dimensionless scaling factors combine rainfall scale, model scale and rainfall input resolution and enable identification of critical rainfall resolution thresholds to achieve a given level of accuracy. Thus, the scaling factors support selection of adequate rainfall resolution to obtain a certain level of accuracy in the calculation of hydrological response.

However, there are still some aspects that need further investigation. Rainfall events measured over the study area should be evaluated: although the assumption of Dutch and British events having the same climatological characteristics, local data need to be evaluated as well. In particular, local rainfall data should be used as input for the model and local discharge measurement should be used as reference when investigating the sensitivity of the hydrological response. Results presented in this paper are

5 related to one specific study case and need to be generalized investigating different study cases. More and different rainfall events and different  catchments 
[revised manuscript text omitted]
).  Temporal percentage of coverage above the selected threshold,  defined as number of time steps above the threshold at each pixel, divided by the total duration of the event (b, e, h, k). Temporal percentage is presented for each rainfall event and the number above each boxplot indicates the total duration of the rainfall event. Cluster dimensions across all time steps per event for the four selected thresholds (c, f, i, l). Blue dots represent the average, green or red lines the median, boxes indicate the first to third quartile and whiskers extend 1.5 times the interquartile range below the first and above the third quartile.

[Figure]

**Figure 4.** Variability of the lag time, depending on the location, for each model (a). The boxplots represent the median (red line), the upper (third quartile) and lower (first quartile) quartile (boxes boundaries), and 1.5 times the interquartile range below the first and above the third quartile (whiskers). Drainage areas corresponding to each location are presented in Table 3(b). Average, median, minimum and maximum value of the lag time as function of $A_d$ for SD2.(b) Fitting power law curves and the power law relation proposed by Berne et al. (2004) are plotted.

[Figure]

**Figure 5.** Peak attenuation ratio $Re_R$ for the 9 rainfall events, as a function  of temporal and spatial rainfall resolution. Symbols indicate the median over the 9 events, solid lines represent the first to the third quartile, dotted lines vary from minimum to maximum. Colours represent different temporal resolutions and markers used for the median indicate different spatial resolutions.

[Figure]

**Figure 6.** Impact of aggregation in space and time on rainfall peak ($Re_R$) and overall pattern ($R_R^2$) for two selected events, as function of sub-catchment size ($A_d$). E4 is a constant low intensity event with low spatial variability. E9 is an example of intermittent event, with a high storm motion velocity. Different colors and symbols indicates different rainfall resolutions used as input. Other events are presented in the supplement material

[Figure]

**Figure 7.** Relative error in peak $Re_Q$ and coefficient of determination $R_Q^2$ for SD2, plotted as function of $A_d$, for the sixteen combinations of rainfall input resolutions. Two different events are presented: E4, a low-intensity constant event, and E9, a multiple peak event.

[Figure]

(a)                                          (b)

**Figure 8.** $Re_Q$ and $R_Q^2$ variability, in relation to model type and rainfall characterized by cluster dimension $S_{Z75}$, for all locations and all combinations of rainfall input resolution. Colours identify the three different models.

[Figure]

**Figure 9.** $R_Q^2$ at Loc2 for different rainfall resolution, plotted against different rainfall characterising scales: spatial (a) and temporal (b) required resolution, Spatial Variability Index (c), dimension of cluster above $Z_{75}$ (d) and $Z_{95}$ (e) and maximum wet period above $Z_{75}$ (f).

[Figure]

**Figure 10.** $Re_Q$ and $R_Q^2$ as function of cluster dimension above $Z_{75}$ and $A_d$. Different colors and symbols indicates different rainfall resolution input.

[Figure]

**Figure 11.** Performance statistic $R_Q^2$ as a function of dimensionless numbers $\theta_S$, $\theta_T$, $\theta_{ST}$, $\alpha_1$, $\alpha_2$, $\alpha_3$. For each parameter all events, rainfall resolutions and locations are plotted.

[Figure]

**Figure 12.** Logarithmic plots of $R_Q^2$ as function of (a) $\alpha_1$, (b) $\alpha_2$ and (c) $\alpha_3$. Different colours indicate different resolutions.

**Table 1.** List of Symbols and Abbreviations

| | | | |
|---|---|---|---|
| Model Characterization | SD1 | | Low resolution semi-distributed model |
| | SD2 | | High resolution semi-distributed model |
| | FD | | Fully distributed model |
| | $A$ | [$L^2$] | Total catchment area |
| | $L_C$ | [L] | Characteristic length of the catchment |
| | $L_C$ | [L] | Characteristic length of the catchment |
| | $L_{RA}$ | [L] | Spatial resolution of the runoff model |
| | $L_S$ | [L] | Sewer length |
| | $t_{lag}$ | [T] | Lag time centroid to centroid |
| Rainfall Resolution | $\Delta s$ | [L] | Spatial resolution used to measure rainfall |
| | $\Delta t$ | [min] | Temporal resolution used to measure rainfall |
| | $N_{tot}$ | [-] | Total number of pixels over the catchment |
| | $d$ | [T] | Rainfall event duration |
| Variogram | $\gamma$ | | Climatological semi - variogram |
| | $n$ | [-] | Number of radar pixel |
| | $R$ | [$L\,T^{-1}$] | Rainfall Rate |
| | $r$ | [L] | Variogram range |
| | $A_r$ | [$L^2$] | Areal average of spatial rainfall structure |
| | $r_c$ | [L] | Characteristic length scale |
| | $\delta s_r$ | [L] | Minimum required spatial resolution |
| | $\delta t_r$ | [T] | Minimum required temporal resolution |
| | $|\bar{v}|$ | [$L\,T^{-1}$] | Storm motion |
| Spatial Variability Index | $I_\sigma$ | [$L\,T^{-1}$] | Spatial variability index |
| | $\sigma_t$ | [$L\,T^{-1}$] | Standard deviation of spatially distributed hourly rainfall |
| | $R_t$ | [$L\,T^{-1}$] | Spatially averaged rainfall intensity per time-step |
| Statistical indicators | $P_{st}$ | [$L\,T^{-1}$] | Rainfall peak of aggregated resolution in space $s$ and time $t$ |
| | $P_{ref}$ | [$L\,T^{-1}$] | Rainfall peak measured at 100 m - 1 min resolution |
| | $Re_R$ | [-] | Peak attenuation ratio |
| | $Re_Q$ | [-] | Relative error on maximum flow peak |
| | $R_R^2$ | [-] | Coefficient of determination for rainfall |
| | $R_Q^2$ | [-] | Coefficient of determination for flow |

**Table 2.** List of Symbols and Abbreviations

[revised manuscript text omitted]